# Object-aware Contrastive Learning for Debiased Scene Representation

**Sangwoo Mo**[*1], **Hyunwoo Kang**[*1], **Kihyuk Sohn**[2], **Chun-Liang Li**[2], **Jinwoo Shin**[1]
[1]KAIST    [2]Google Cloud AI
{swmo,hyunwookang,jinwoos}@kaist.ac.kr, {kihyuks,chunliang}@google.com

## Abstract

Contrastive self-supervised learning has shown impressive results in learning visual representations from unlabeled images by enforcing invariance against different data augmentations. However, the learned representations are often contextually biased to the spurious scene correlations of different objects or object and background, which may harm their generalization on the downstream tasks. To tackle the issue, we develop a novel object-aware contrastive learning framework that first (a) localizes objects in a self-supervised manner and then (b) debias scene correlations via appropriate data augmentations considering the inferred object locations. For (a), we propose the contrastive class activation map (ContraCAM), which finds the most discriminative regions (e.g., objects) in the image compared to the other images using the contrastively trained models. We further improve the ContraCAM to detect multiple objects and entire shapes via an iterative refinement procedure. For (b), we introduce two data augmentations based on ContraCAM, object-aware random crop and background mixup, which reduce contextual and background biases during contrastive self-supervised learning, respectively. Our experiments demonstrate the effectiveness of our representation learning framework, particularly when trained under multi-object images or evaluated under the background (and distribution) shifted images.[1]

## 1 Introduction

Self-supervised learning of visual representations from unlabeled images is a fundamental task of machine learning, which establishes various applications including object recognition [1, 2], reinforcement learning [3, 4], out-of-distribution detection [5, 6], and multimodal learning [7, 8]. Recently, contrastive learning [1, 2, 9–15] has shown remarkable advances along this line. The idea is to learn invariant representations by attracting the different views (e.g., augmentations) of the same instance (i.e., positives) while contrasting different instances (i.e., negatives).[2]

Despite the success of contrastive learning on various downstream tasks [16], they still suffer from the generalization issue due to the unique features of the training datasets [17–19] or the choice of data augmentations [19–21]. In particular, the co-occurrence of different objects and background in randomly cropped patches (i.e., positives) leads the model to suffer from the *scene bias*. For example, Figure 1a presents two types of the scene bias: the positive pairs contain different objects (e.g., giraffe and zebra), and the patches contain adjacent object and background (e.g., zebra and safari). Specifically, the co-occurrence of different objects is called contextual bias [22], and that of object and background is called background bias [23]. Attracting the patches in contrastive learning

---

[*]Equal contribution

[1]Code is available at https://github.com/alinlab/object-aware-contrastive.

[2]Some recent works (e.g., [14, 15]) attract the positives without contrasting the negatives. While we mainly focus on contrastive learning with negatives, our method is also applicable to the positive-only methods.

35th Conference on Neural Information Processing Systems (NeurIPS 2021).

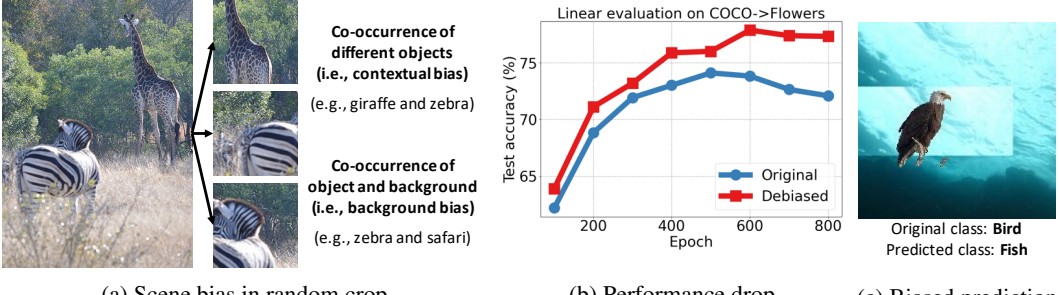

(a) Scene bias in random crop     (b) Performance drop     (c) Biased prediction

Figure 1: Scene bias issue (a) and its negative effects on contrastive learning. (b) Linear evaluation [24] of the original MoCov2 [1] and our debiased method, trained and evaluated on the COCO [25] and Flowers [26] datasets, respectively, using the ResNet-50 architecture [27]. The vanilla MoCov2 often loses its discriminative power as training goes as it entangles different objects, while the debiased model stably improves the classification performance. (c) Prediction of MoCov2 on an image from the Background Challenge [23]. The vanilla MoCov2 makes decisions from the background instead of the object, leading to biased prediction on background-shifted images.

makes the features of correlated objects and background indistinguishable, which may harm their generalization (Figure 1b) because of being prone to biases (Figure 1c).

**Contribution.** We develop a novel object-aware contrastive learning framework that mitigates the scene bias and improves the generalization of learned representation. The key to success is the proposed *contrastive class activation map* (ContraCAM), a simple yet effective self-supervised object localization method by contrasting other images to find the most discriminate regions in the image. We leverage the ContraCAM to create new types of positives and negatives. First, we introduce two data augmentations for constructing the *positive* sample-pairs of contrastive learning: *object-aware random crop* and *background mixup* that reduce contextual and background biases, respectively. Second, by equipping ContraCAM with an iterative refinement procedure, we extend it to detect multiple objects and entire shapes, which allows us to generate masked images as effective *negatives*.

We demonstrate that the proposed method can improve two representative contrastive (or positive-only) representation learning schemes, MoCov2 [28] and BYOL [14], by reducing contextual and background biases as well as learning object-centric representation. In particular, we improve:

- The representation learning under multi-object images, evaluated on the COCO [25] dataset, boosting the performance on the downstream tasks, e.g., classification and detection.

- The generalizability of the learned representation on the background shifts, i.e., objects appear in the unusual background (e.g., fish on the ground), evaluated on the Background Challenge [23].

- The generalizability of the learned representation on the distribution shifts, particularly for the shape-biased, e.g., ImageNet-Sketch [29], and corrupted, e.g., ImageNet-C [30] datasets.

Furthermore, ContraCAM shows comparable results with the state-of-the-art unsupervised localization method (and also with the supervised classifier CAM) while being simple.

## 2 Object-aware Contrastive Learning

We first briefly review contrastive learning in Section 2.1. We then introduce our object localization and debiased contrastive learning methods in Section 2.2 and Section 2.3, respectively.

### 2.1 Contrastive learning

Contrastive self-supervised learning aims to learn an encoder $f(\cdot)$ that extracts a useful representation from an unlabeled image $x$ by attracting similar sample $x^+$ (i.e., positives) and dispelling dissimilar samples $\{x_i^-\}$ (i.e., negatives). In particular, instance discrimination [10] defines the same samples of different data augmentations (e.g., random crop) as the positives and different samples as negatives.

Formally, contrastive learning maximizes the contrastive score:

$$s_{\texttt{con}}(x; x^+, \{x_n^-\}) := \log \frac{\exp(\text{sim}(z(x), \bar{z}(x^+))/\tau)}{\exp(\text{sim}(z(x), \bar{z}(x^+))/\tau) + \sum_{x_n^-} \exp(\text{sim}(z(x), \bar{z}(x_n^-))/\tau)}, \quad (1)$$

where $z(\cdot)$ and $\bar{z}(\cdot)$ are the output and target functions wrapping the representation $f(x)$ for use, $\text{sim}(\cdot, \cdot)$ denotes the cosine similarity, and $\tau$ is a temperature hyperparameter. The specific form of $z(\cdot), \bar{z}(\cdot)$ depends on the method. For example, MoCov2 [28] sets $z(\cdot) = g(f(\cdot)), \bar{z}(\cdot) = g_m(f_m(\cdot))$ where $g(\cdot)$ is a projector network to indirectly match the feature $f(x)$ and $f_m(\cdot), g_m(\cdot)$ are the momentum version of the encoder and projectors. On the other hand, BYOL [14] sets $z(\cdot) = h(g(f(\cdot))), \bar{z}(\cdot) = g_m(f_m(\cdot))$, where $h(\cdot)$ is an additional predictor network to avoid collapse of the features because it only maximizes the similarity score $s_{\texttt{sim}}(x; x^+) := \text{sim}(z(x), \bar{z}(x^+))$ [14, 15].

**Scene bias in contrastive learning.** Despite the success of contrastive learning, they often suffer from the *scene bias*: entangling representations of co-occurring (but different) objects, i.e., contextual bias [22], or adjacent object and background, i.e., background bias [23], by attracting the randomly cropped patches reflecting the correlations (Figure 1a). The scene bias harms the performance (Figure 1b) and generalization of the learned representations on distribution shifts (Figure 1c). To tackle the issue, we propose object-aware data augmentations for debiased contrastive learning (Section 2.3) utilizing the object locations inferred from the contrastively trained models (Section 2.2).

## 2.2  ContraCAM: Unsupervised object localization via contrastive learning

We aim to find the most discriminative region in an image, such as objects for scene images, compared to the other images. To this end, we extend the (gradient-based) class activation map (CAM) [31, 32], originally used to find the salient regions for the prediction of classifiers. Our proposed method, *contrastive class activation map* (ContraCAM), has two differences from the classifier CAM. First, we use the contrastive score instead of the softmax probability. Second, we discard the negative signals from the similar objects in the negative batch since they cancel out the positive signals and hinder the localization, which is crucial as shown in Table 1 and Appendix C.1).

Following the classifier CAM, we define the saliency map as the weighted sum of spatial activations (e.g., penultimate feature before pooling), where the weight of each activation is given by the importance, the sum of gradients, of the activation for the score function. Formally, let $\mathbf{A} := [A_{ij}^k]$ be a spatial activation of an image $x$ where $1 \le i \le H, 1 \le j \le W, 1 \le k \le K$ denote the index of row, column, and channel, and $H, W, K$ denote the height, width, and channel size of the activation. Given a batch of samples $\mathcal{B}$, we define the score function of the sample $x$ as the contrastive score $s_{\texttt{con}}$ in Eq. (1) using the sample $x$ itself as a positive[3] and the remaining samples $\mathcal{B} \setminus x$ as negatives. Then, the weight of the $k$-th activation $\alpha_k$ and the CAM mask $\texttt{CAM} := [\texttt{CAM}_{ij}] \in [0, 1]^{H \times W}$ are:

$$\texttt{CAM}_{ij} = \texttt{Normalize}\left(\texttt{ReLU}\left(\sum_k \alpha_k A_{ij}^k\right)\right), \quad \alpha_k = \texttt{ReLU}\left(\frac{1}{HW}\sum_{i,j} \frac{\partial s_{\texttt{con}}(x; x, \mathcal{B} \setminus x)}{\partial A_{i,j}^k}\right), \quad (2)$$

where $\texttt{Normalize}(x) := \frac{x - \min x}{\max x - \min x}$ is a normalization function that maps the elements to $[0, 1]$. We highlight the differences from the classifier CAM with the red color. Note that the $\texttt{ReLU}$ used to compute $\alpha_k$ in Eq. (2) discards the negative signals. The negative signal removal trick also slightly improves the classifier CAM [33] but much effective for the ContraCAM.

We further improve the ContraCAM to detect multiple objects and entire shapes with an iterative refinement procedure [34]: cover the salient regions of the image with the (reverse of) current CAM, predict new CAM from the masked image, and aggregate them (see Figure 2). It expands the CAM regions since the new CAM from the masked image detects the unmasked regions. Here, we additionally provide the masked images in the batch (parelly computed) as the negatives: they are better negatives by removing the possibly existing similar objects. Also, we use the original image $x$ as the positive to highlight the undetected objects. Formally, let $\texttt{CAM}^t$ be the CAM of iteration $t$ and $\overline{\texttt{CAM}}^t := [\overline{\texttt{CAM}}_{ij}^t] = [\max_{l \le t} \texttt{CAM}_{ij}^l]$ be the aggregated CAM mask. Also, let $x^t$ be the image softly masked by the (reverse of) current aggregated mask, i.e., $x^t := (1 - \overline{\texttt{CAM}}^{t-1}) \odot x$ for $t \ge 2$

---

[3]It does not affect the score but is defined for the notation consistency with the iterative extension.

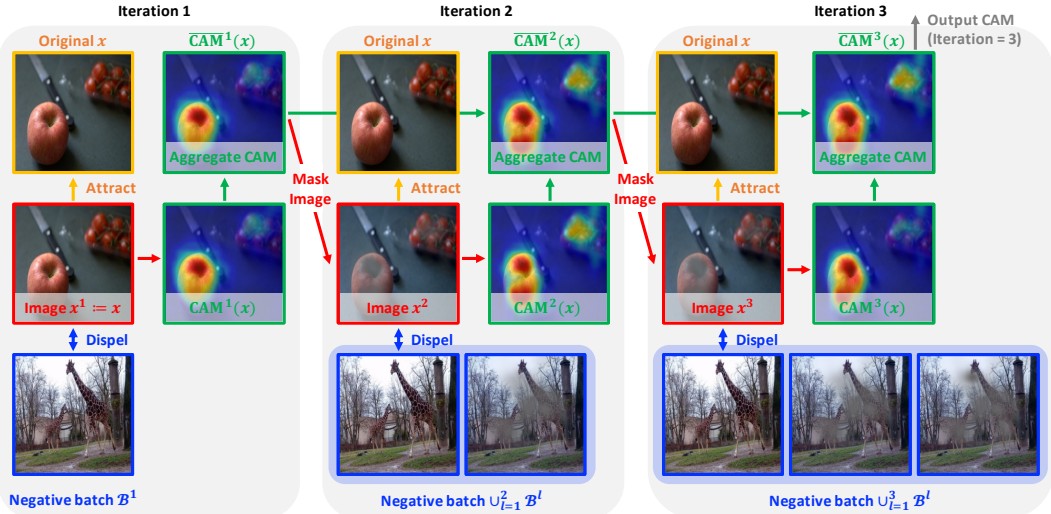

Figure 2: Visual illustration of the Iterative ContraCAM ($T = 3$) procedure.

and $x^1 = x$ where $\odot$ denotes an element-wise product, and $\mathcal{B}^t := \{x^t_n\}$ be the batch of the masked images. Then, we define the score function for iteration $t$ as:

$$s^t_{\text{con}}(x) := s_{\text{con}}(x^t; x, \cup_{l \leq t}(\mathcal{B}^l \setminus x^l)). \tag{3}$$

We substitute the contrastive score $s_{\text{con}}$ in Eq. (2) with the $s^t_{\text{con}}$ in Eq. (3) to compute the CAM of iteration $t$, and use the final aggregated mask after $T$ iterations. We remark that the CAM results are not sensitive to the number of iterations $T$ if it is large enough; CAM converges to the stationary value since soft masking $x^t$ regularizes the CAM not to be keep expanded (see Appendix C.2). We provide the pseudo-code of the entire Iterative ContraCAM procedure in Appendix A.

Note that contrastive learning was known to be ineffective at localizing objects [35] with standard saliency methods (using a classifier on top of the learned representation) since attracting the randomly cropped patches makes the model look at the entire scene. To our best knowledge, we are the first to extend the CAM for the self-supervised setting, relaxing the assumption of class labels. Selvaraju et al. [36] considered CAM for contrastive learning, but their purpose was to regularize CAM to be similar to the ground-truth masks (or predicted by pre-trained models) and used the similarity of the image and the masked image (by ground-truth masks) as the score function of CAM.

### 2.3 Object-aware augmentations for debiased contrastive learning

We propose two data augmentations for contrastive learning that reduce contextual and background biases, respectively, utilizing the object locations inferred by ContraCAM. Both augmentations are applied to the *positive* samples before other augmentations; thus, it is applicable for both contrastive learning (e.g., MoCov2 [28]) and positive-only methods (e.g., BYOL [14]).

**Reducing contextual bias.** We first tackle the contextual bias of contrastive learning, i.e., entangling the features of different objects. To tackle the issue, we propose a data augmentation named *object-aware random crop*, which restricts the random crop around a single object and avoids the attraction of different objects. To this end, we first extract the (possibly multiple or none) bounding boxes of the image from the binarized mask[4] of the ContraCAM. We then crop the image around the box, randomly chosen from the boxes, before applying other augmentations (e.g., random crop). Here, we apply augmentations (to produce positives) to the same cropped box; thus, the patches are restricted in the same box. Technically, it only requires a few line addition of code:

```
if len(boxes) > 0: # can be empty
    box = random.choice(boxes)
    image = image.crop(box)
# apply other augmentations (e.g., random crop)
```

---

[4]Threshold the mask or apply a post-processing method, e.g., conditional random field (CRF) [37].

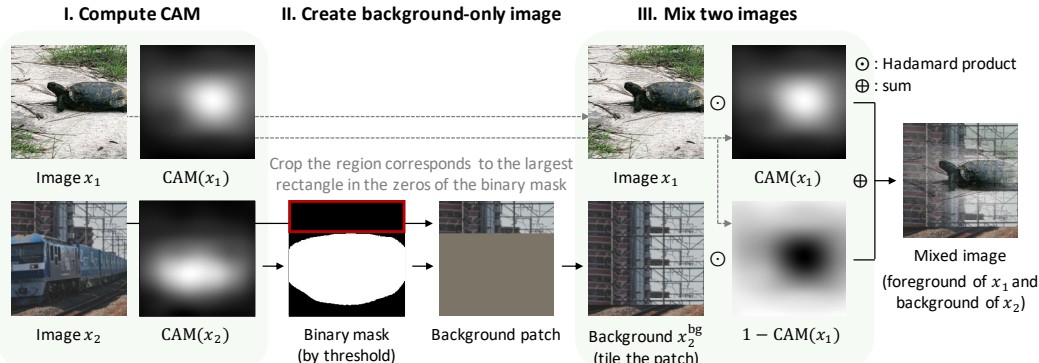

Figure 3: Visual illustration of the background mixup procedure.

Purushwalkam and Gupta [19] considered a similar approach using ground-truth bounding boxes applied on MoCov2. However, we found that cropping around the ground-truth boxes often harms contrastive learning (see Table 4). This is because some objects (e.g., small ones) in ground-truth boxes are hard to discriminate (as negatives), making contrastive learning hard to optimize. In contrast, the ContraCAM produces more discriminative boxes, often outperforming the ground-truth boxes (see Appendix D.1). Note that the positive-only methods do not suffer from the issue: both ground-truth and ContraCAM boxes work well. On the other hand, Selvaraju et al. [36] used a pre-trained segmentation model to constrain the patches to contain objects. It partly resolves the false positive issue by avoiding the attraction of background-only patches but does not prevent the patches with different objects; in contrast, the object-aware random crop avoids both cases.

**Reducing background bias.** We then tackle the background bias of contrastive learning, i.e., entangling the features of adjacent object and background. To this end, we propose a data augmentation named *background mixup*, which substitutes the background of an image with other backgrounds. Intuitively, the positive samples share the objects but have different backgrounds, thus reducing the background bias. Formally, background mixup blends an image $x_1$ and a background-only image $x_2^{\text{bg}}$ (generated from an image $x_2$) using the ContraCAM of image $x_1$ as a weight, i.e.,

$$x_1^{\text{bg-mix}} := \text{CAM}(x_1) \odot x_1 + (1 - \text{CAM}(x_1)) \odot x_2^{\text{bg}}, \tag{4}$$

where $\odot$ denotes an element-wise product. Here, the background-only image $x_2^{\text{bg}}$ is generated by tiling the background patch of the image $x_2$ inferred by the ContraCAM. Precisely, we choose the largest rectangle in the zeros of the binarized CAM mask for the region of the background patch. The overall procedure of the background mixup is illustrated in Figure 3.

Prior works considered the background bias for contrastive learning [35, 38] but used a pre-trained segmentation model and copy-and-pasted the objects to the background-only images using binary masks. We also tested the copy-and-paste version with the binarized CAM, but the soft version in Eq. (4) performed better (see Appendix E.1); one should consider the confidence of the soft masks since they are inaccurate. Furthermore, the background mixup improves the generalization on distribution shifts, e.g., shape-biased [29, 39, 40] and corrupted [30] datasets (see Table 8). Remark that the background mixup often outperforms the Mixup [41] and CutMix [42] applied for contrastive learning [43]. Intuitively, the background mixup can be viewed as a saliency-guided extension [44, 45] of mixup but not mixing the targets (positives), since the mixed patch should be only considered as the positive of the patch sharing foreground, not the one sharing background.

## 3  Experiments

We first verify the localization performance of ContraCAM in Section 3.1. We then demonstrate the efficacy of our debiased contrastive learning: object-aware random crop improves the training under multi-object images by reducing contextual bias in Section 3.2, and background mixup improves generalization on background and distribution shifts by reducing background bias in Section 3.3.

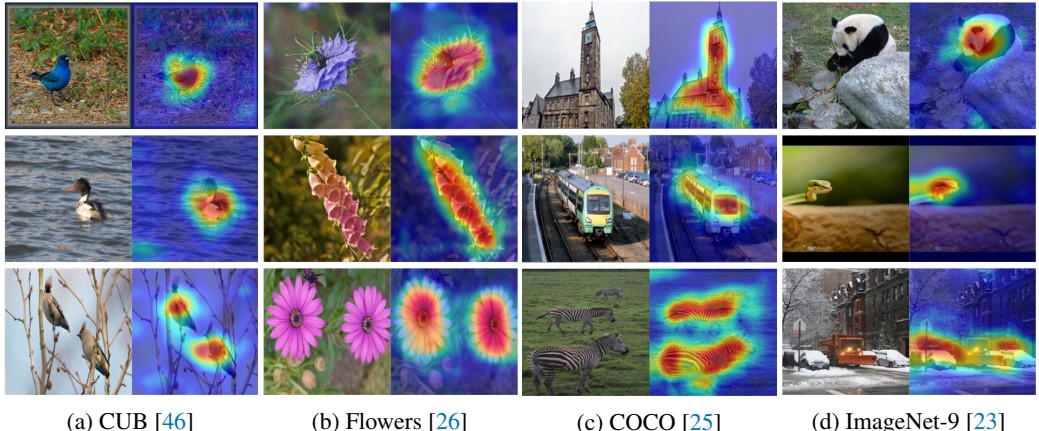

| (a) CUB [46] | (b) Flowers [26] | (c) COCO [25] | (d) ImageNet-9 [23] |

Figure 4: Visualization of the ContraCAM results on various image datasets.

Table 1: Mask mIoU of unsupervised object localization methods. Bold denotes the best results.

| Method | CUB | Flowers | COCO | ImageNet-9 |
|---|---|---|---|---|
| ReDO [47] | 0.426 | 0.764 | 0.286 | 0.416 |
| ContraCAM w/o negative signal removal | 0.287 | 0.555 | 0.242 | 0.361 |
| ContraCAM (ours) | **0.460** | **0.776** | **0.319** | **0.427** |

**Common setup.** We apply our method on two representative contrastive (or positive-only) learning models: MoCov2 [28] and BYOL [14], under the ResNet-18 and ResNet-50 architectures [27]. We train the models for 800 epochs on COCO [25] and ImageNet-9 [23], and 2,000 epochs on CUB [46] and Flowers [26] datasets with batch size 256. For object localization experiments, we train the vanilla MoCov2 and BYOL on each dataset and compute the CAM masks. For representation learning experiments, we first train the vanilla MoCov2 and BYOL to pre-compute the CAM masks (and corresponding bounding boxes); then, we retrain MoCov2 and BYOL, applying our proposed augmentations using the fixed pre-computed masks (and boxes). Here, we retrain the models from scratch to make the training budgets fair. We also retrained (i.e., third iteration) the model using the CAM masks from our debiased models but did not see the gain (see Appendix D.6). We follow the default hyperparameters of MoCov2 and BYOL, except the smaller minimum random crop scale of 0.08 (instead of the original 0.2) since it performed better, especially for the multi-object images. We run a single trial for contextual bias and three trials for background bias experiments.

We use the penultimate spatial activations to compute the CAM results. At inference, we follow the protocol of [48] that doubly expands the resolution of the activations to detect the smaller objects through decreasing the stride of the convolutional layer in the final residual block. Since it produces the smaller masks, we use more iterations (e.g., 10) for the Iterative ContraCAM. Here, we apply the conditional random field (CRF) using the default hyperparameters from the pydensecrf library [49] to produce segmentation masks and use the opencv [50] library to extract bounding boxes. We use a single iteration of the ContraCAM without the expansion trick for background bias results; it is sufficient for single instance images. Here, we binarize the masks with a threshold of 0.2 to produce background-only images. We provide the further implementation details in Appendix B.

**Computation time.** The training of the baseline models on the COCO (∼100,000 samples) dataset takes ∼1.5 days on 4 GPUs and ∼3 days on 8 GPUs for ResNet-18 and ResNet-50 architectures, respectively, using a single machine with 8 GeForce RTX 2080 Ti GPUs; proportional to the number of samples and training epochs for other cases. The inference of ContraCAM takes a few minutes for the entire training dataset, and generating the boxes using CRF takes dozens of minutes. Using the pre-computed masks and boxes, our method only slightly increases the training time.

## 3.1 Unsupervised object localization

We check the performance of our proposed self-supervised object localization method, ContraCAM. Figure 4 shows the examples of the ContraCAM on various image datasets, including CUB, Flowers,

Table 2: Mask mIoU of the classifier CAM using a supervised model and ContraCAM using MoCov2, where both models are solely trained on the target dataset and evaluated on the same dataset.

| Training | Inference | CUB | Flowers | ImageNet-9 |
|---|---|---|---|---|
| Supervised | Classifier CAM | 0.451 | 0.633 | 0.509 |
| MoCov2 | ContraCAM (ours) | 0.460 | 0.776 | 0.427 |

Table 3: MaxBoxAccV2 of the classifier CAM and ContraCAM using the ImageNet-trained classifier and MoCov2. We report the localization results on the trained (ImageNet) and unseen datasets.

| Training | Inference | ImageNet | CUB | Flowers | VOC | OpenImages |
|---|---|---|---|---|---|---|
| Supervised | Classifier CAM | 55.95 | 55.52 | 76.87 | 53.88 | 48.01 |
| MoCov2 | ContraCAM (ours) | 55.88 | 64.07 | 75.64 | 59.40 | 49.89 |

COCO, and ImageNet-9 datasets. ContraCAM even detects multiple objects in the image. We also quantitatively compare ContraCAM with the state-of-the-art unsupervised object localization method, ReDo [47]. Table 1 shows that the ContraCAM is comparable with ReDO, in terms of the the mask mean intersection-over-unions (mIoUs). One can also see that the negative signal removal, i.e., ReLU in Eq. (1), is a critical to the performance (see Appendix C.1 for the visual examples).

We also compare the localization performance of ContraCAM (using MoCov2) and classifier CAM (using a supervised model). Table 2 shows the results where all models are solely trained from the target dataset and evaluated on the same dataset. Interestingly, ContraCAM outperforms the classifier CAM on CUB and Flowers. We conjecture this is because CUB and Flowers have few training samples; the supervised classifier is prone to overfitting. On the other hand, Table 3 shows the results on the transfer setting, i.e., the models are trained on the ImageNet [51] using the ResNet-50 architecture. We use the publicly available supervised classifier [52] and MoCov2, and follow the MaxBoxAccV2 evaluation protocol [48]. The ContraCAM often outperforms the classifier CAM, especially for the unseen images (e.g., CUB). This is because the classifiers project out the features unrelated to the target classes, losing their generalizability on the out-of-class samples.

We provide additional analysis and results in Appendix C. Appendix C.2 shows the ablation study on the number of iterations of ContraCAM. One needs a sufficient number of iterations since too few iterations often detect subregions. Since ContraCAM converges to the stationary values for more iterations, we simply choose 10 for all datasets. Appendix C.3 shows the effects of the negative batch of ContraCAM. Since ContraCAM finds the most discriminative regions compared to the negative batch, one needs to choose the negative batch different from the target image. Using a few randomly sampled images is sufficient. Appendix C.4 provides additional comparison of ContraCAM and classifier CAM. Finally, Appendix C.5 provides a comparison with the gradient-based saliency methods [53, 54] using the same contrastive score. CAM gives better localization results.

## 3.2 Reducing contextual bias: Representation learning from multi-object images

We demonstrate the effectiveness of the object-aware random crop (OA-Crop) for representation learning under multi-object images by reducing contextual bias. To this end, we train MoCov2 and BYOL on the COCO dataset, comparing them with the models that applied the OA-Crop using the ground-truth (GT) bounding boxes or inferred ones from the ContraCAM.

We first compare the linear evaluation [24], test accuracy of a linear classifier trained on top of the learned representation, in Table 4. We report the results on the COCO-Crop, i.e., the objects in the COCO dataset cropped by the GT boxes, CIFAR-10 and CIFAR-100 [55], CUB, Flowers, Food [56], and Pets [57] datasets. OA-Crop significantly improves the linear evaluation of MoCov2 and BYOL for all tested cases. Somewhat interestingly, OA-Crop using the ContraCAM boxes even outperforms the GT boxes for MoCov2 under the ResNet-50 architecture. This is because the GT boxes often contain objects hard to discriminate (e.g., small objects), making contrastive learning hard to optimize; in contrast, ContraCAM finds more distinct objects. Note that BYOL does not suffer from this issue and performs well with both boxes. See Appendix D.1 for the detailed discussion.

We also compare the detection (and segmentation) performance measured by mean average precision (AP), an area under the precision-recall curve of the bounding boxes (or segmentation masks),

Table 4: Linear evaluation (%) of MoCov2 and BYOL on various image classification tasks, trained with the original image (-) or object-aware random crop (OA-Crop) using the ContraCAM (CAM) or ground-truth (GT) bounding boxes from the COCO dataset. Gray lines denote the usage of GT boxes, blue and red brackets denote the gain and loss of OA-Crop compared to the original image.

| Model | Network | OA-Crop | Test dataset | | | | | | |
|---|---|---|---|---|---|---|---|---|---|
| | | | COCO-Crop | CIFAR10 | CIFAR100 | CUB | Flowers | Food | Pets |
| MoCov2 | ResNet-50 | - | 74.30 | 77.58 | 53.26 | 22.90 | 72.09 | 59.70 | 59.25 |
| MoCov2 | ResNet-50 | CAM | 76.37 (+2.07) | 84.10 (+6.52) | 62.72 (+9.46) | 25.46 (+2.56) | 77.33 (+5.24) | 62.01 (+2.31) | 60.97 (+1.72) |
| MoCov2 | ResNet-50 | GT | 76.44 (+2.14) | 84.03 (+6.45) | 62.81 (+9.55) | 22.59 (-0.31) | 75.09 (+3.00) | 57.47 (-2.23) | 57.67 (-1.58) |
| BYOL | ResNet-50 | - | 73.36 | 76.62 | 51.79 | 21.95 | 73.77 | 59.49 | 60.72 |
| BYOL | ResNet-50 | CAM | 74.92 (+1.56) | 82.79 (+6.17) | 61.13 (+9.34) | 24.34 (+2.39) | 77.83 (+4.06) | 61.83 (+2.34) | 61.27 (+0.55) |
| BYOL | ResNet-50 | GT | 80.69 (+7.33) | 85.92 (+9.30) | 65.06 (+13.27) | 28.68 (+6.73) | 77.95 (+4.18) | 64.63 (+5.14) | 65.69 (+4.97) |
| MoCov2 | ResNet-18 | - | 67.38 | 66.83 | 41.85 | 15.36 | 58.81 | 45.88 | 45.37 |
| MoCov2 | ResNet-18 | CAM | 69.92 (+2.54) | 76.73 (+9.90) | 53.25 (+11.40) | 16.26 (+0.90) | 64.77 (+5.96) | 48.56 (+2.68) | 47.37 (+2.00) |
| MoCov2 | ResNet-18 | GT | 71.60 (+4.22) | 77.99 (+11.16) | 53.32 (+11.47) | 18.19 (+2.83) | 65.43 (+6.62) | 46.41 (+0.53) | 48.68 (+3.31) |
| BYOL | ResNet-18 | - | 67.74 | 67.82 | 41.96 | 17.24 | 64.79 | 49.58 | 52.90 |
| BYOL | ResNet-18 | CAM | 70.85 (+3.11) | 77.37 (+9.55) | 54.79 (+12.83) | 18.24 (+1.00) | 70.56 (+5.77) | 53.16 (+3.58) | 54.27 (+1.37) |
| BYOL | ResNet-18 | GT | 76.59 (+8.85) | 81.23 (+13.41) | 58.11 (+16.15) | 22.99 (+5.75) | 73.25 (+8.46) | 55.33 (+5.75) | 59.80 (+6.90) |

Table 5: Mean AP (%) of MoCov2 and BYOL fine-tuned on the COCO detection and segmentation tasks, following the setting of the table above, using the ResNet-50 architecture.

| | MoCov2 | | | BYOL | | |
|---|---|---|---|---|---|---|
| | Baseline | OA-Crop (CAM) | OA-Crop (GT) | Baseline | OA-Crop (CAM) | OA-Crop (GT) |
| COCO Detection | 36.34 | 36.60 (+0.26) | 35.73 (-0.61) | 35.11 | 35.63 (+0.52) | 35.05 (-0.06) |
| COCO Segmentation | 31.95 | 32.37 (+0.42) | 31.47 (-0.48) | 31.10 | 31.39 (+0.29) | 31.12 (+0.02) |

Table 6: Test accuracy (%) of a linear classifier evaluated on various distribution-shifted datasets, following the setting of the table above, using the ResNet-50 architecture.

| Model | Crop | Test dataset | | | | |
|---|---|---|---|---|---|---|
| | | ImageNet-9 | ImageNet-Sketch-9 | Stylized-ImageNet-9 | ImageNet-R-9 | ImageNet-C-9 |
| MoCov2 | Baseline | 84.67 | 41.44 | 18.94 | 32.40 | 26.08 |
| MoCov2 | OA-Crop (CAM) | 84.54 (-0.13) | 43.11 (+1.68) | 20.50 (+1.56) | 32.35 (-0.05) | 27.85 (+1.77) |
| MoCov2 | OA-Crop (GT) | 82.49 (-2.18) | 46.85 (+5.42) | 22.18 (+3.24) | 33.68 (+1.28) | 26.81 (+0.73) |
| BYOL | Baseline | 84.07 | 44.28 | 17.91 | 32.13 | 27.51 |
| BYOL | OA-Crop (CAM) | 84.67 (+0.60) | 45.05 (+0.77) | 20.21 (+2.29) | 32.64 (+0.51) | 28.70 (+1.19) |
| BYOL | OA-Crop (GT) | 86.72 (+2.65) | 51.52 (+7.25) | 22.95 (+5.03) | 36.28 (+4.15) | 31.65 (+4.14) |

on the COCO detection and segmentation tasks in Table 5. Here, we fine-tune the MoCov2 and BYOL models using the ResNet-50 architecture. Remark that OA-Crop using the ContraCAM boxes outperforms the baselines, while the GT boxes are on par or worse. This is because the GT boxes solely focus on the objects while ContraCAM also catches the salient scene information.

In addition, we present the generalization performance of learned representations under the distribution shifts in Table 6. To this end, we evaluate the models trained on the COCO dataset to various 9 superclass (370 classes) subsets of ImageNet, whose details will be elaborated in the next section. ImageNet-9 contains natural images like COCO, but other datasets contain distribution-shifted (e.g., shape-biased or corrupted) images. Note that OA-Crop performs on par with the vanilla MoCov2 and BYOL on the original ImageNet-9 but performs better on the distribution-shifted dataset. It verifies that the OA-Crop improves the generalizability of the learned representation.

We provide additional analysis and results in Appendix D. Appendix D.2 provides an additional analysis that OA-Crop indeed reduces the contextual bias. Specifically, the representation learned from OA-Crop shows better separation between the co-occurring objects, giraffe and zebra. Appendix D.3 provides the comparison with the supervised representation, learned by Faster R-CNN [58] and Mask R-CNN [59], using ground-truth bounding boxes or segmentation masks. OA-Crop significantly reduces the gap between self-supervised and supervised representation. Appendix D.4 presents the class-wise accuracy on CIFAR10 that OA-Crop consistently improves the accuracy over all classes. Appendix D.5 presents the linear evaluation performance of MoCov2 and BYOL trained on a 10% subset of ImageNet for readers comparing with the results with the ImageNet-trained models.

Table 7: Test accuracy (%) of a linear classifier evaluated on the Background Challenge [23], both backbone and classifier are trained under the ORIGINAL dataset. The backbone is trained from the original image (Baseline), background mixup using ContraCAM (BG-Mixup (CAM)), or hard background mixing using ground-truth masks (BG-HardMix (GT)), under the ResNet-18 architecture. Blue (or red) arrows imply higher (or lower) is better. Subscripts denote standard deviation.

| | MoCov2 | | | BYOL | | |
|---|---|---|---|---|---|---|
| Dataset | Baseline | BG-Mixup (CAM) | BG-HardMix (GT) | Baseline | BG-Mixup (CAM) | BG-HardMix (GT) |
| Original ↑ | $89.17_{\pm0.49}$ | $90.73_{\pm0.05}$ (+1.56) | $89.69_{\pm0.14}$ (+0.52) | $87.30_{\pm0.61}$ | $89.30_{\pm0.02}$ (+2.00) | $90.95_{\pm0.33}$ (+3.65) |
| Only-BG-B ↓ | $31.29_{\pm2.46}$ | $29.60_{\pm0.89}$ (-1.69) | $26.44_{\pm1.63}$ (-4.85) | $25.59_{\pm0.78}$ | $25.70_{\pm3.46}$ (+0.11) | $27.28_{\pm0.04}$ (+1.69) |
| Only-BG-T ↓ | $44.91_{\pm0.16}$ | $41.95_{\pm0.38}$ (-2.96) | $40.11_{\pm0.58}$ (-4.80) | $42.83_{\pm0.51}$ | $39.94_{\pm0.52}$ (-2.89) | $41.16_{\pm0.17}$ (-1.67) |
| Only-FG ↑ | $63.62_{\pm4.71}$ | $70.55_{\pm1.71}$ (+6.93) | $72.68_{\pm0.69}$ (+9.06) | $61.04_{\pm0.94}$ | $67.53_{\pm0.30}$ (+6.49) | $72.63_{\pm1.13}$ (+11.59) |
| Mixed-Same ↑ | $80.98_{\pm0.34}$ | $84.13_{\pm0.33}$ (+3.15) | $84.48_{\pm0.17}$ (+3.50) | $79.30_{\pm0.31}$ | $81.28_{\pm0.53}$ (+1.98) | $84.94_{\pm0.47}$ (+5.64) |
| Mixed-Rand ↑ | $60.34_{\pm0.66}$ | $66.89_{\pm0.54}$ (+6.55) | $71.95_{\pm0.54}$ (+11.61) | $58.03_{\pm0.85}$ | $63.83_{\pm0.53}$ (+5.80) | $70.51_{\pm0.33}$ (+12.48) |
| Mixed-Next ↑ | $55.50_{\pm0.71}$ | $63.64_{\pm0.41}$ (+8.14) | $70.25_{\pm0.14}$ (+14.75) | $53.35_{\pm0.36}$ | $63.05_{\pm3.54}$ (+9.70) | $66.81_{\pm0.08}$ (+13.46) |
| BG-Gap ↓ | $20.64_{\pm0.36}$ | $17.24_{\pm0.31}$ (-3.40) | $12.53_{\pm0.69}$ (-8.11) | $21.27_{\pm0.64}$ | $17.45_{\pm0.15}$ (-3.82) | $14.44_{\pm0.56}$ (-6.83) |

Table 8: Test accuracy (%) of a linear classifier evaluated on various distribution-shifted datasets, following the training of the table above, additionally comparing with Mixup [41] and CutMix [42].

| Model | Augmentation | Test dataset | | | |
|---|---|---|---|---|---|
| | | ImageNet-Sketch-9 | Stylized-ImageNet-9 | ImageNet-R-9 | ImageNet-C-9 |
| MoCov2 | Baseline | $46.70_{\pm0.67}$ | $25.66_{\pm0.54}$ | $37.51_{\pm0.80}$ | $31.82_{\pm0.40}$ |
| MoCov2 | +Mixup [41] | $51.18_{\pm0.88}$ (+4.48) | $32.36_{\pm0.12}$ (+6.70) | $41.00_{\pm0.12}$ (+3.49) | $40.15_{\pm2.07}$ (+8.33) |
| MoCov2 | +CutMix [42] | $45.92_{\pm0.88}$ (-0.78) | $26.46_{\pm0.68}$ (+0.80) | $37.07_{\pm0.31}$ (-0.44) | $32.29_{\pm0.60}$ (+0.47) |
| MoCov2 | +BG-Mixup (ours) | $\mathbf{52.15}_{\pm0.93}$ (+5.45) | $\mathbf{33.36}_{\pm0.61}$ (+7.70) | $\mathbf{41.50}_{\pm0.45}$ (+3.99) | $\mathbf{44.39}_{\pm0.89}$ (+12.57) |
| BYOL | Baseline | $45.15_{\pm1.12}$ | $23.80_{\pm0.45}$ | $36.21_{\pm0.31}$ | $28.62_{\pm0.06}$ |
| BYOL | +Mixup [41] | $50.12_{\pm1.61}$ (+1.97) | $\mathbf{28.11}_{\pm1.15}$ (+4.31) | $37.90_{\pm0.44}$ (+1.69) | $32.48_{\pm0.55}$ (+3.86) |
| BYOL | +CutMix [42] | $46.07_{\pm0.05}$ (+1.39) | $23.98_{\pm0.05}$ (+0.18) | $35.43_{\pm0.44}$ (-0.78) | $29.68_{\pm0.39}$ (+1.06) |
| BYOL | +BG-Mixup (ours) | $\mathbf{52.40}_{\pm0.70}$ (+7.25) | $27.01_{\pm0.74}$ (+3.21) | $\mathbf{39.62}_{\pm0.21}$ (+3.41) | $\mathbf{33.83}_{\pm0.28}$ (+5.21) |

### 3.3 Reducing background bias: Generalization on background and distribution shifts

We demonstrate the effectiveness of the background mixup (BG-Mixup) for the generalization of the learned representations on background and distribution shifts by reducing background bias and learning object-centric representation. To this end, we train MoCov2 and BYOL (and BG-Mixup upon them) on the ORIGINAL dataset from the Background Challenge [23], a 9 superclass (370 classes) subset of the ImageNet [51]. We then train a linear classifier on top of the learned representation using the ORIGINAL dataset. Here, we evaluate the classifier on the Background Challenge datasets for the background shift results, and the corresponding 9 superclasses of the ImageNet-Sketch [29], Stylized-ImageNet [39], ImageNet-R [40], and ImageNet-C [30] datasets, denoted by putting '-9' at the suffix of the dataset names, for the distribution shift results (see Appendix B.3 for details).

We additionally compare BG-Mixp with the hard background mixing (i.e., copy-and-paste) using ground-truth masks (BG-HardMix (GT)) for the background shift experiments, and Mixup [41] and CutMix [42] (following the training procedure of [43]) for the distribution shift experiments. We also tested the BG-HardMix using the binarized CAM but did not work well (see Appendix E.1). On the other hand, the BG-Mixup often makes contrastive learning hard to be optimized by producing hard positives; thus, we apply BG-Mix with probability $p_{\texttt{mix}} < 1$, independently applied on the patches. We tested $p_{\texttt{mix}} \in \{0.2, 0.3, 0.4, 0.5\}$ and choose $p_{\texttt{mix}} = 0.4$ for MoCov2 and $p_{\texttt{mix}} = 0.3$ for BYOL. Note that MoCov2 permits the higher $p_{\texttt{mix}}$, since finding the closest sample from the (finite) batch is easier than clustering infinitely many samples (see Appendix E.2 for details).

Table 7 presents the results on background shifts: BG-Mixup improves the predictions on the object-focused datasets (e.g., MIXED-RAND) while regularizing the background-focused datasets (e.g., ONLY-BG-T). Table 8 presents the results on distribution shifts: BG-Mixup mostly outperforms the Mixup and the CutMix. We also provide the BG-HardMix (GT) results on distribution shifts in Appendix E.3 and the mixup results on background shifts in Appendix E.4. The superiority of BG-Mix on both background and distribution shifts shows that its merits come from both object-centric learning via reducing background and the saliency-guided input interpolation. In addition, we provide the corruption-wise classification results on ImageNet-9-C in Appendix E.5, and additional distribution shifts results on ObjectNet [60] and SI-Score [61] in Appendix E.6.

# 4 Related work

**Contrastive learning.** Contrastive learning (or positive-only method) [1, 2, 14] is the state-of-the-art method for visual representation learning, which incorporates the prior knowledge of invariance over the data augmentations. However, they suffer from an inherent problem of matching false positives from random crop augmentation. We tackle this scene bias issue and improve the quality of learned representation. Note that prior work considering the scene bias for contrastive learning [19, 35, 36, 38] assumed the ground-truth object annotations or pre-trained segmentation models, undermining the motivation of self-supervised learning to reduce such supervision. In contrast, we propose a fully self-supervised framework of object localization and debiased contrastive learning. Several works [62, 63] consider an object-aware approach for video representation learning, but their motivation was to attract the objects of different temporal views and require a pretrained object detector.

**Bias in visual representation.** The bias (or shortcut) in neural networks [64] have got significant attention recently, pointing out the unintended over-reliance on texture [39], background [23], adversarial features [65], or conspicuous inputs [66]. Numerous works have thus attempted to remove such biases, particularly in an unsupervised manner [29, 67, 68]. Our work also lies on this line: we evoke the scene bias issue of self-supervised representation learning and propose an unsupervised debiasing method. Our work would be a step towards an unbiased, robust visual representation.

**Unsupervised object localization.** The deep-learning-based unsupervised object localization methods can be categorized as follow. (a) The generative-based [47, 69, 70] approaches train a generative model that disentangles the objects and background by enforcing the object-perturbed image to be considered as real. (b) The noisy-ensemble [71–73] approaches train a model using handcrafted predictions as noisy targets. Despite the training is unsupervised, they initialize the weights with the supervised model. (c) Voynov et al. [74] manually finds the 'salient direction' from the noise (latent) of the ImageNet-trained BigGAN [75]. Besides, scene decomposition (e.g., [76]) aims at a more ambitious goal: fully decompose the objects and background, but currently not scale to the complex images. To our best knowledge, the generative-based approach is the state-of-the-art method for fully unsupervised scenarios. Our proposed ContraCAM could be an alternative in this direction.

**Class activation map.** Class activation map [31, 32] has been used for the weakly-supervised object localization (WSOL), inferring the pixel- (or object-) level annotations using class labels. Specifically, classifier CAM finds the regions that are most salient for the classifier score. ContraCAM further expands its applicability from weakly-supervised to unsupervised object localization by utilizing the contrastive score instead of the classifier score. We think ContraCAM will raise new interesting research questions, e.g., one could adopt the techniques from CAM to the ContraCAM.

# 5 Conclusion and Discussion

We proposed the ContraCAM, a simple and effective self-supervised object localization method using the contrastively trained models. We then introduced two data augmentations upon the ContraCAM that reduce scene bias and improve the quality of the learned representations for contrastive learning. We remark that the scene bias is more severe for the uncurated images; our work would be a step towards strong self-supervised learning under real-world scenarios [77, 78].

**Limitations.** Since the ContraCAM finds the most salient regions, it can differ from the desiderata of the users, e.g., the ContraCAM detects both the birds and branches in the CUB [46] dataset, but one may only want to detect the birds. Also, though the ContraCAM identifies the disjoint objects, it is hard to separate the occluded objects. Incorporating the prior knowledge of the objects and designing a more careful method to disentangle objects would be an interesting future direction.

**Potential negative impacts.** Our proposed framework enforces the model to focus on the "objects", or the salient regions, to disentangle the relations of the objects and background. However, ContraCAM may over-rely on the conspicuous objects and the derived data augmentation strategy by ContraCAM could potentially incur imbalanced performance across different objects. We remark that the biases in datasets and models cannot be entirely eliminated without carefully designed guidelines. While we empirically observe our proposed learning strategies mitigate contextual and background biases on certain object types, we still need a closer look at the models, interactively correcting them.

## Acknowledgements

This work was partly supported by Institute of Information & Communications Technology Planning & Evaluation (IITP) grant funded by the Korea government (MSIT) (No.2019-0-00075, Artificial Intelligence Graduate School Program (KAIST); No. 2019-0-01396, Development of framework for analyzing, detecting, mitigating of bias in AI model and training data; No.2017-0-01779, A machine learning and statistical inference framework for explainable artificial intelligence), and partly by the Defense Challengeable Future Technology Program of the Agency for Defense Development, Republic of Korea. We thank Jihoon Tack, Jongjin Park, and Sihyun Yu for their valuable comments.

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
