# A Algorithms

**Algorithm 1** PyTorch-style pseudo-code for the Iterative ContraCAM.

```
# input: image (b, c, h, w)

masked_image = image # initial: original image

queues = []
for i in n_iters:
    feature = get_features(masked_image) # spatial features
    feature.requires_grad = True
    output = get_projection(feature) # projection outputs

    if i == 0:
        key = output.detach() # original images
    queues.append(output.detach()) # masked images

    score = contrastive_score(output, key, queues) # See Algorithm 2
    cam = compute_cam(feature, score, size=(h, w)) # See Algorithm 3

    mask = max(mask, cam) if i > 0 else cam # union over iterations
    masked_image = image * (1 - mask) + mask_color * mask # soft mask

return mask
```

**Algorithm 2** PyTorch-style pseudo-code for the contrastive score.

```
# input: query (b,d), key (b,d), queues List[(b,d)]

pos = einsum('nc,nc->n', [query, key])
neg = cat([einsum('nc,kc->nk', [query, q]) * (1 - query.size(0))
          for q in queues], dim=1)

score = (pos.exp().sum(dim=1) / neg.exp().sum(dim=1)).log()
return score
```

**Algorithm 3** PyTorch-style pseudo-code for the Class Activation Map (CAM).

```
# input: feature (b,c,h,w), score (b), size=(H,W)

grad = autograd.grad(score.sum(), feature)[0]

weight = adaptive_avg_pool2d(grad, output_size=(1, 1))
weight = weight.clamp_min(0) # clamp negative weights

cam = sum(weight * feature, dim=1, keepdim=True).detach() # weighted sum
cam = interpolate(cam, size=(H,W)) # scale-up to image size
cam = normalize(relu(cam)) # normalize to [0,1]
return cam
```

# B  Implementation details

We build our code upon the PyTorch [52] and PyTorch Lightning[5] library. Further implementation details and additional libraries for each experiment are stated in the remaining subsections.

## B.1  Implementation details for object localization results

We train MoCov2 under the ResNet-18 architecture on CUB, Flowers, COCO, and ImageNet-9 datasets for the segmentation results. We train the models with batch size 256, COCO, and ImageNet-9 for 800 epochs and CUB and Flowers for 2,000 epochs since the latter has few samples. We follow the augmentations of He et al. [1]: color jitter with strength (0.4,0.4,0.4,0.1), random grayscale with probability 0.2, and Gaussian blur with kernel size 23 and standard deviation sampled from (0.1,2.0) with probability 0.5; except random crop patches with size (0.08,1.0) instead of the original (0.2,1.0) as it performed better for images with small objects. We use a learning rate of 0.03 with a cosine annealing schedule. These training configurations are applied for all experiments.

We apply the expansion trick [48]: doubly expand the resolution of penultimate spatial activations by decreasing the stride of the convolutional layer in the final residual block to detect small objects with CAM. Note that we only apply this trick at inference time and do not change the training; namely, the model is trained with the original $7 \times 7$ resolution but inferred with the expanded $14 \times 14$ of the spatial activations. We also tried training the models using the modified $14 \times 14$ resolution but did not see much gain. We run ten iterations for the Iterative ContraCAM and apply the conditional random field following the default hyperparameters[6] from the pydensecrf library [49]. We report the mask mean intersection-over-union (mIoU) between the predicted and ground-truth segmentation masks.

For the comparison of the classifier CAM and ContraCAM, we use the publicly available supervised classifier[7] and MoCov2[8] trained on the ImageNet dataset under the ResNet-50 architecture. Here, we do not apply the expansion trick and run a single iteration for the ContraCAM. We report the MaxBoxAccV2 [48]: averages the ratios of the bounding boxes whose mean intersection-over-unions (mIoUs) are larger than 30%, 50%, and 70% where the boxes for each mIoU percentages are generated by the CAM binarized by the optimal thresholds, on the ImageNet, CUB, Flowers, VOC, and OpenImages dataset following the official evaluation code.[9] Recall that we report the transfer performance of the predicted CAMs from the ImageNet-trained models for these experiments.

## B.2  Implementation details for contextual bias results

We train MoCov2 and BYOL under the ResNet-18 and ResNet-50 architectures on the COCO dataset for 800 epochs with batch size 256. We extract the bounding boxes from the binarized CAM masks using the `findContours` function in the OpenCV library [50]. We compute the boxes with MoCov2 trained on ResNet-18 and ResNet-50 architectures and use them for the debiased MoCov2 and BYOL using the same architectures. We found that giving some margin for the boxes slightly improves the performance by observing more object boundaries. Specifically, we expand the boxes with 20% of margins (width for left-and-right and height for up-and-down) found from the experiments using the ground-truth boxes and use the same margins for the CAM boxes. We also remove the small boxes, specifically smaller than 1% of the image size, to remove vague low-resolution objects.

We follow the linear evaluation scheme of Chen et al. [2]: train a $\ell_2$-regularized multinomial logistic regression classifier on top of the pre-computed representation using the L-BFGS [79] optimizer. We compute the representation with the center cropped images and choose the $\ell_2$-regularization parameter from $(10^{-6}, 10^5)$ spaced with a range of 45 logarithmically. We evaluate the transfer performance on the COCO-Crop (crop objects of the COCO dataset with 20% of margins), CIFAR-10, CIFAR-100, CUB, Flowers, Food, and Pets datasets using the linear classifier trained and tested on each dataset. For detection experiments, we follow the fine-tuning configuration of He et al. [1] evaluated on the COCO dataset. We use the Detectron[10] library for the detection experiments.

---

[5] https://github.com/PyTorchLightning/pytorch-lightning
[6] https://github.com/lucasb-eyer/pydensecrf
[7] https://pytorch.org/vision/stable/models.html
[8] https://github.com/facebookresearch/moco
[9] https://github.com/clovaai/wsolevaluation
[10] https://github.com/facebookresearch/Detectron

### B.3 Implementation details for background bias results

We provide the visual examples of the Background Challenge [23] in Figure 5 and distribution-shifted datasets of ImageNet [51]: ImageNet-Sketch [29], Stylized-ImageNet [39], ImageNet-R [40], and ImageNet-C [30] datasets in Figure 6. We train the models on the ImageNet-9 [23], i.e., the ORIGINAL dataset of the Background Challenge, which contains 9 superclass (370 class) of the full ImageNet for both background and distribution shifts experiments. Thus, we use the the corresponding 9 superclass subsets of the distribution-shifted datasets, denoted by putting '-9' at the suffix of the dataset names.

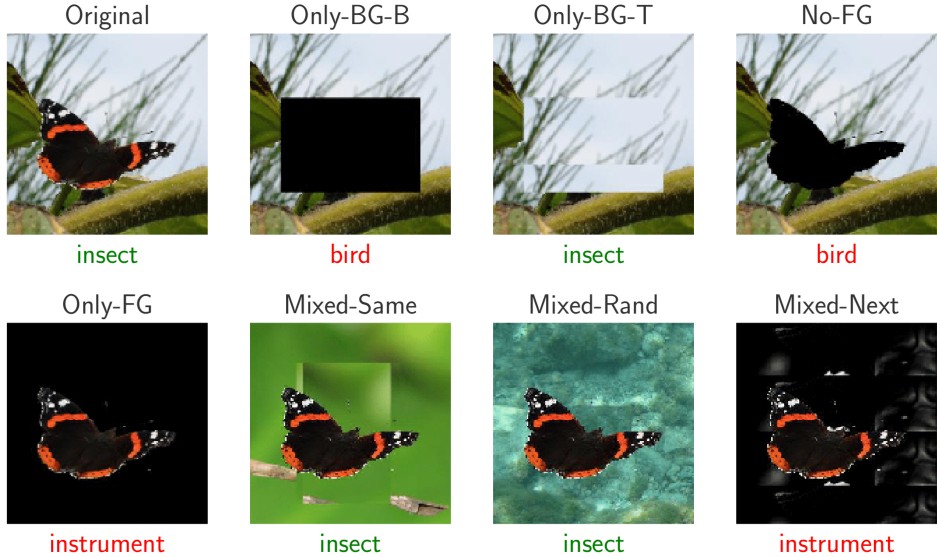

Figure 5: Visual examples of the Background Challenge [23]. Image from the original paper.

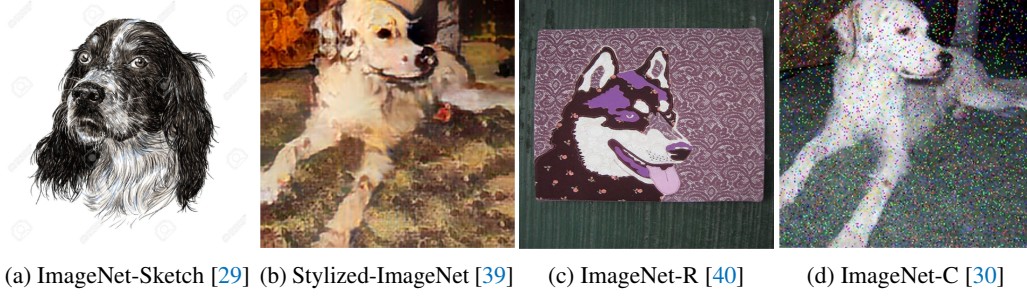

(a) ImageNet-Sketch [29] (b) Stylized-ImageNet [39]  (c) ImageNet-R [40]  (d) ImageNet-C [30]

Figure 6: Visual examples of the distribution-shifted datasets of ImageNet [51] for 'dog' class.

We train MoCov2 and BYOL under the ResNet-18 architecture on the ORIGINAL dataset of the Background Challenge for 800 epochs with batch size 256. We use the ContraCAM masks from MoCov2 to train debiased MoCov2 and BYOL for debiased BYOL. We threshold the CAM values with a threshold of 0.2 to find the largest contour, find the largest rectangle outside the contour to create the background patch and tile it for the background-only image. We train a linear classifier on the ORIGINAL dataset and evaluate test accuracy on the Background Challenge and distribution-shifted ImageNet 9 superclass subsets for the background and distribution shift results.

# C   Additional localization results

## C.1   Visualization of ContraCAM without negative signal removal

Figure 7 shows the examples of ContraCAM without negative signal removal. The negative signals from similar objects in different images disturb the localization results by canceling positive signals; spread in random locations. Therefore, removing these signals improves the localization results.

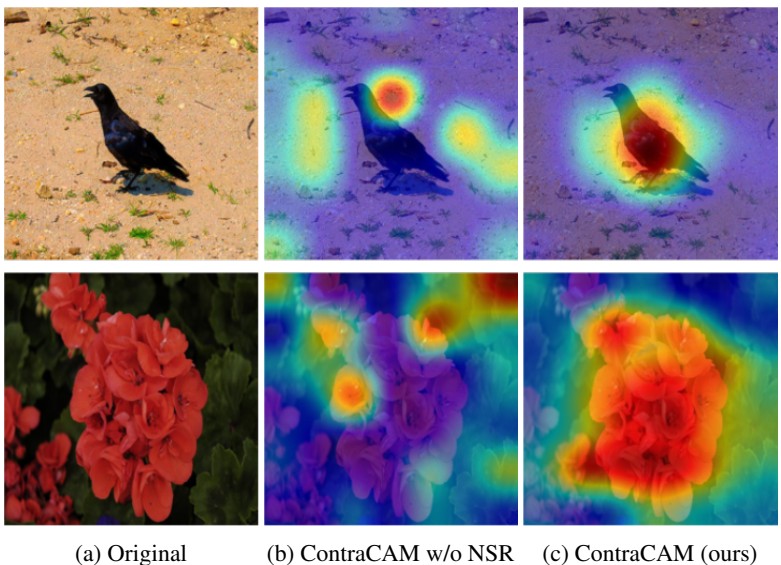

(a) Original          (b) ContraCAM w/o NSR     (c) ContraCAM (ours)

Figure 7: Visualization of ContraCAM without negative signal removal (NSR).

## C.2   Ablation study on the number of iterations

We present the ablation study on the number of iterations for the ContraCAM in Table 9. One needs a sufficient number of iterations (e.g., 5) since too small numbers of iterations often detect subregions or miss some objects. Also, note that the CAM shows stable results for large numbers (e.g., 20) of iterations and converges to some stationary values, though it slightly harms the best value of 10.

Table 9: Mask mIoU of ContraCAM with various number of iterations.

| Iteration | CUB | Flowers | COCO | ImageNet-9 |
|---|---|---|---|---|
| 1 | 0.249 | 0.374 | 0.081 | 0.090 |
| 2 | 0.402 | 0.662 | 0.182 | 0.220 |
| 3 | 0.447 | 0.738 | 0.256 | 0.328 |
| 5 | **0.461** | 0.753 | 0.308 | 0.417 |
| 10 | 0.460 | **0.776** | **0.319** | **0.427** |
| 20 | 0.458 | 0.737 | 0.318 | 0.419 |

## C.3 Ablation study on the choice of negative batch

We study the effects of the negative batch for the ContraCAM. Recall that the ContraCAM finds the most discriminative regions compared to the negative batch; it assumes that images have similar backgrounds but different objects. For a sanity check, we construct a negative batch containing similar objects. Figure 8 shows an example of the ContraCAM using a giraffe-only and random batch as the negatives. ContraCAM highlights background when compared to the giraffe-only batch.

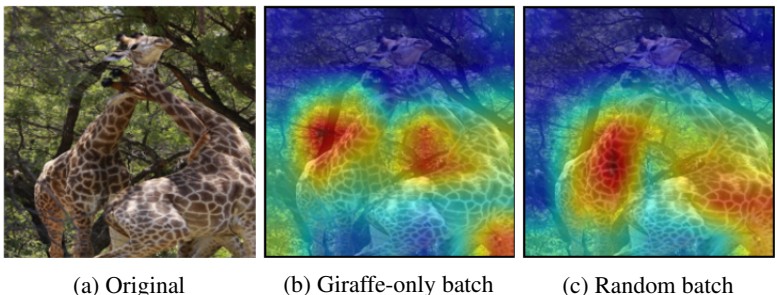

(a) Original      (b) Giraffe-only batch      (c) Random batch

Figure 8: Visualization of the similar-objects and random negative batches.

However, the pathological selection of the negative batch rarely occurs in practice; using a small number of random samples can alleviate the issue. Table 10 shows the effects of the negative batch size for the ContraCAM. Using a small batch (e.g., of size 4) almost match the performance of the larger batch (e.g., of size 64). We use the random batch of size 64 for all our experiments.

Table 10: Mask mIoU of ContraCAM with various negative batch sizes.

| Batch size | CUB | Flowers | COCO | ImageNet-9 |
|:----------:|:-----:|:-------:|:-----:|:----------:|
| 4 | 0.451 | 0.731 | 0.315 | 0.428 |
| 16 | 0.455 | 0.731 | 0.317 | **0.429** |
| 64 | **0.460** | **0.776** | **0.319** | 0.427 |

## C.4 Comparison with the Classifier CAM

We compare the ContraCAM and classifier CAM under the publicly available supervised classifier and MoCov2 trained on the ImageNet dataset. Somewhat interestingly, the ContraCAM often outperforms the classifier CAM on the transfer setting, e.g., when transferred to the CUB dataset, as shown in Figure 9. This is because some samples of the CUB dataset are out-of-class of the ImageNet, and the classifier fails to understand the important features unrelated to the original classes.

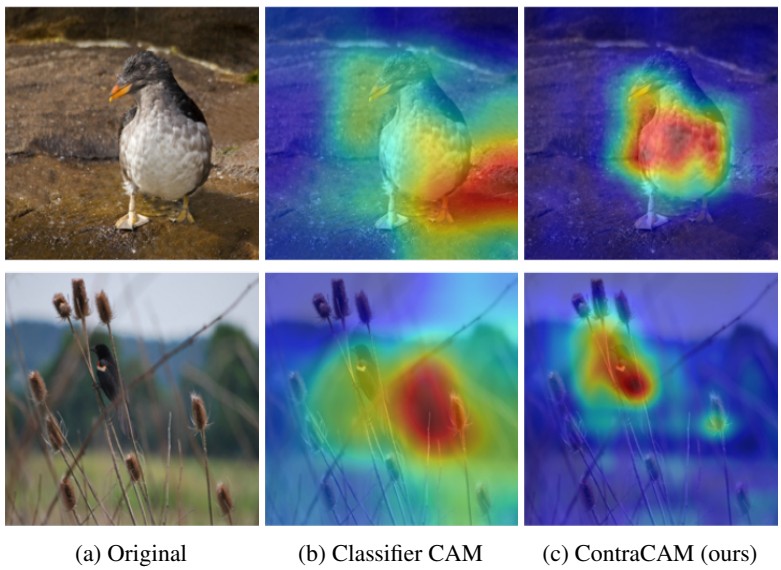

|              (a) Original              |              (b) Classifier CAM              |              (c) ContraCAM (ours)              |

Figure 9: Visualization of the Classifier CAM and ContraCAM.

To check whether the superiority of the ContraCAM comes from the score function or better backbone, we also train a linear classifier on top of the MoCov2 backbone using the ImageNet dataset and test the classifier CAM. Table 9 shows that the ContraCAM on MoCov2 even outperforms the classifier CAM on the same backbone for the ImageNet to CUB transfer scenario.

On the other hand, the table shows that the double expansion trick [48] of the resolution of penultimate spatial activations is more effective for MoCov2 while degrading the supervised classifier; MoCov2 is trained with stronger augmentations, making CAM robust to the modification of the architecture. Thus, we only apply the expansion trick for the MoCov2 results in Table 2.

Table 11: MaxBoxAccV2 of the Classifier CAM and ContraCAM using the supervised classifier and MoCov2 trained on the ImageNet dataset under the ResNet-50 architecture. Res×2 denotes the usage of the double expansion trick [48] of the resolution of penultimate spatial activations.

| Model      | Method           | Res×2 | ImageNet | CUB   | Flowers | VOC   | OpenImages |
|------------|------------------|-------|----------|-------|---------|-------|------------|
| Supervised | Classifier CAM   |       | 55.95    | 55.52 | 76.87   | 53.88 | 48.01      |
| Supervised | Classifier CAM   | ✓     | 55.01    | 43.23 | 73.31   | 52.27 | 47.23      |
| MoCov2     | Classifier CAM   |       | 57.79    | 63.84 | 74.29   | 59.45 | 51.99      |
| MoCov2     | Classifier CAM   | ✓     | 60.04    | 62.87 | 78.01   | 61.03 | 53.06      |
| MoCov2     | ContraCAM (ours) |       | 54.57    | 60.33 | 74.29   | 58.64 | 48.84      |
| MoCov2     | ContraCAM (ours) | ✓     | 55.88    | 64.07 | 75.64   | 59.40 | 49.89      |

## C.5 Comparison with the gradient-based saliency methods

We compare the ContraCAM and gradient-based saliency methods using the contrastive score Eq. (1). All methods use the same score function but only differ from localization: the weighted sum of activations (i.e., CAM) or directly propagate the gradients to the input space (i.e., gradient-based saliencies). We choose two representative gradient-based saliency methods: Integrated Gradients (IntGrad) [53] and SmoothGrad [54], which ensembles multiple gradients for better saliency detection. Specifically, IntGrad ensembles the gradients of the linear interpolation of the image and the zero image, and SmoothGrad ensembles the gradients of the image added by random Gaussian noises. We average ten gradients, either interpolation or Gaussian noises, for both methods.

Figure 10 and Table 12 present the visual examples and quantitative results measured by MaxBox-AccV2, respectively. The gradient-based saliencies provide sparse points as outputs, which can be hard to aggregate as segmentation masks. In contrast, ContraCAM provides smooth maps which are more interpretable and easily used for applications, e.g., post-process to bounding boxes. Furthermore, the gradient-based saliencies detect larger regions than ContraCAM. We think it is due to the negative signals: unlike ContraCAM, it is non-trivial to remove them for the gradient-based saliencies.

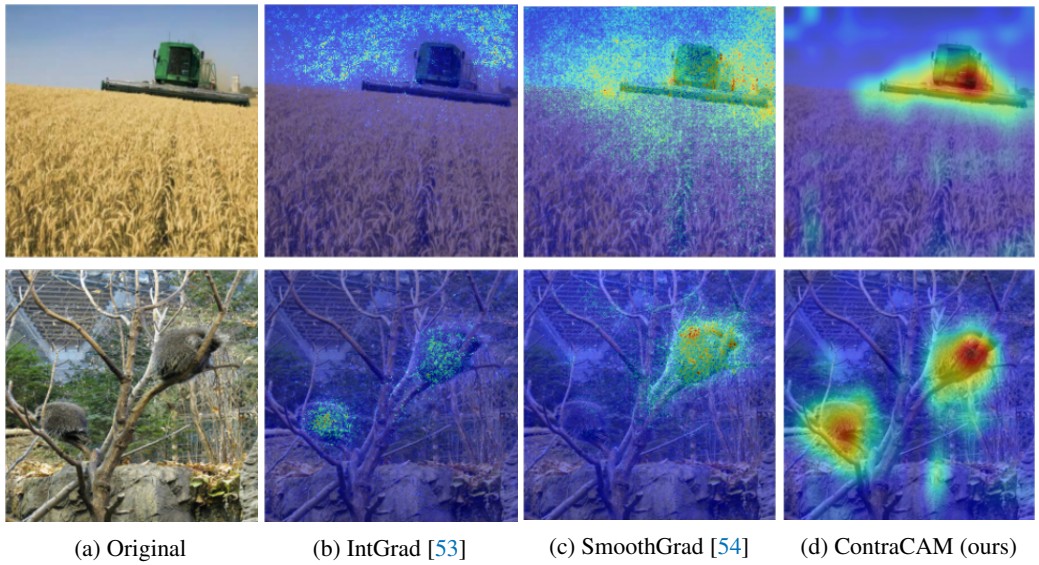

| (a) Original | (b) IntGrad [53] | (c) SmoothGrad [54] | (d) ContraCAM (ours) |

Figure 10: Visualization of various saliency methods using the contrastive score Eq. (1).

Table 12: MaxBoxAccV2 of various saliency methods using the contrastive score Eq. (1). We compute the saliencies from the MoCov2 trained on the ImageNet dataset under the ResNet-50 architecture.

| Method | ImageNet | CUB | Flowers | VOC | OpenImages |
|---|---|---|---|---|---|
| IntGrad [53] | 48.40 | 35.44 | 70.73 | 48.52 | 49.48 |
| SmoothGrad [54] | 51.70 | 51.50 | 72.83 | 57.26 | 48.67 |
| ContraCAM (ours) | **55.88** | **64.07** | **75.64** | **59.40** | **49.89** |

# D Additional contextual bias results

## D.1 Hard negative issue in MoCov2

We found that MoCov2 trained with the object-aware random crop (OA-Crop) using ground-truth (GT) bounding boxes does not perform well, often worse than the original image (Baseline). This is because the contrastive learning objective is hard to optimize and unstable during training for the OA-Crop (GT), as shown in Figure 11. In contrast, OA-Crop using the ContraCAM boxes is much stable, yet it is a little harder to optimize than the original image.

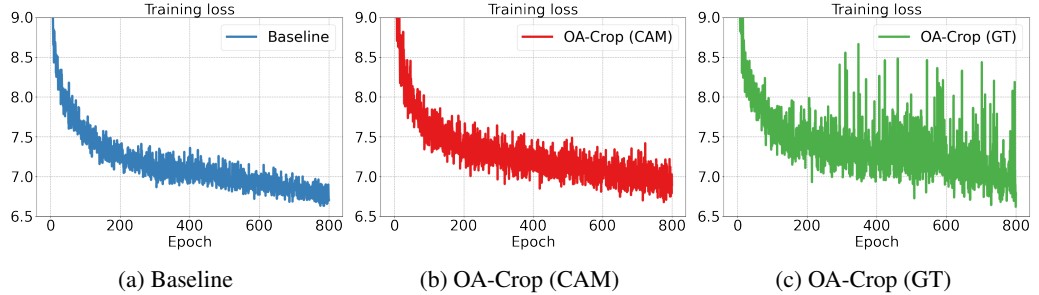

| (a) Baseline | (b) OA-Crop (CAM) | (c) OA-Crop (GT) |

Figure 11: Training loss curve of MoCov2 trained under the COCO dataset.

The reason behind this phenomenon is that the ground-truth boxes often contain objects that are hard to distinguish from each other, i.e., hard negatives for contrastive learning. In contrast, ContraCAM finds more discriminative objects as defined in Eq. (1). Figure 12 shows the histogram of the number of ContraCAM and ground-truth boxes, and Figure 13 shows a visual example of them. ContraCAM finds the most recognizable 1∼3 objects from the full ground-truth boxes.

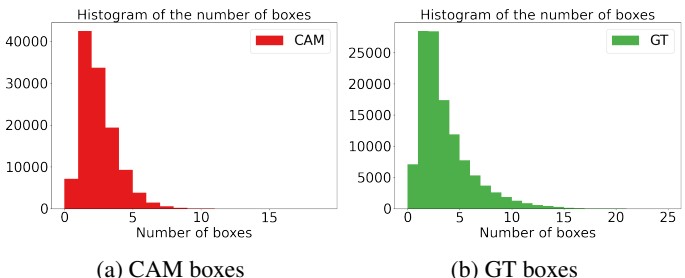

| (a) CAM boxes | (b) GT boxes |

Figure 12: Histogram of the number of ContraCAM and ground-truth boxes.

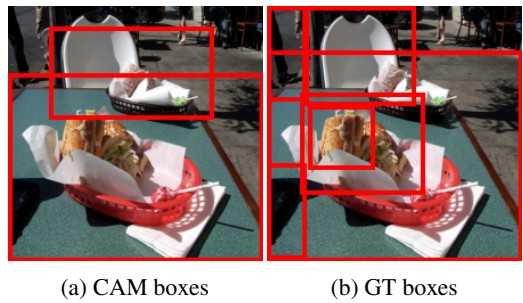

| (a) CAM boxes | (b) GT boxes |

Figure 13: Visualization of the ContraCAM and ground-truth boxes.

## D.2 Analysis on the contextual bias

We analyze whether the object-aware random crop (OA-Crop) actually relieves the contextual bias. To verify this, we visualize the embeddings of correlated classes under the original MoCov2 and the debiased model using the OA-Crop with the ContraCAM boxes. Specifically, we choose giraffe and zebra, which frequently co-occurs in the safari scene (see Figure 1a). Figure 14 shows the t-SNE [80] visualization of the giraffe and zebra embeddings of the original and debiased models. The debiased OA-Crop (CAM) model less entangles the features of giraffe and zebra. However, even the debiased model using the ground-truth boxes, i.e., OA-Crop (GT), does not perfectly disentangle the features; since the bounding boxes often contain nearby or occluded objects.

We also quantitatively measure the contextual bias of the models in Table 13. Specifically, we compute the average minimum $\ell_2$-distance of the features, i.e.,

$$\frac{1}{|\mathcal{X}|} \sum_{x \in \mathcal{X}} \min_{y \in \mathcal{Y}} d(x, y) + \frac{1}{|\mathcal{Y}|} \sum_{y \in \mathcal{Y}} \min_{x \in \mathcal{X}} d(x, y), \tag{5}$$

where $\mathcal{X}, \mathcal{Y} \subset \mathcal{R}^m$ are the penultimate embeddings of each class (giraffe and zebra) and $d$ denotes a $\ell_2$-distance function, under the MoCov2 using the ResNet-50 architecture. The model trained by OA-Crop (CAM) has a larger distance between the embeddings than the original image.

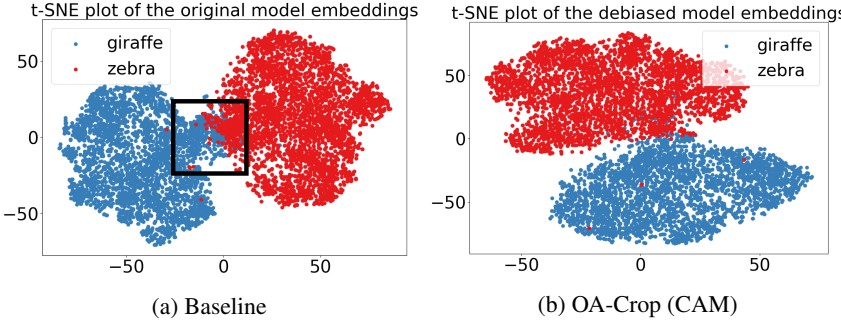

(a) Baseline            (b) OA-Crop (CAM)

Figure 14: t-SNE [80] visualization of the giraffe and zebra embeddings.

Table 13: Average minimum $\ell_2$-distance of giraffe and zebra embeddings.

| Baseline | OA-Crop (CAM) | OA-Crop (GT) |
|----------|---------------|--------------|
| 0.2441   | 0.2790        | 0.3824       |

To further verify that the contextual bias harms the discriminability, we report the classification error of co-occurring classes (giraffe vs. zebra) over epochs. Upon the fixed representation, we compute the 5 seed average of 1-shot binary classification error. The classification error of vanilla MoCov2 increases for later epochs while the object-aware random crop shows consistent results.

Table 14: Error rate (%) of the binary classifier of giraffe vs. zebra over epochs.

| Epoch    | 100  | 200 | 300 | 400 | 500 | 600 | 700 | 800 |
|----------|------|-----|-----|-----|-----|-----|-----|-----|
| MoCov2   | 22.5 | 1.9 | 3.8 | 2.7 | 7.7 | 4.6 | 6.6 | 8.5 |
| +OA-Crop | 1.6  | 1.7 | 2.4 | 2.4 | 5.2 | 2.9 | 1.6 | 1.3 |

## D.3 Comparison with supervised models

We provide the linear evaluation and detection/segmentation results of supervised models. Specificlaly, we consider two representative supervised learning model: Faster R-CNN [58] and Mask R-CNN [59], which are trained on bounding boxes and instance segmentations, respectively. We use the publicly available PyTorch models[11] trained on the COCO dataset using the ResNet-50 architecture. We use the pretrained weights for detection/segmentation experiments (Table 16) and trained a linear classifier upon the pretrained weights for linear evaluation experiments (Table 15).

Table 15 and Table 16 show that the supervised Faster R-CNN and Mask R-CNN learns better representation than the self-supervised models. However, BYOL trained with ground-truth object boxes matches the supervised models' linear evaluation performance, implying the self-supervised methods' potentials. While ContraCAM significantly improves the vanilla MoCov2/BYOL, it would be an interesting future direction to reduce the gap between the supervised models further.

Table 15: Linear evaluation (%) of MoCov2 and BYOL under the ResNet-50 architecture, following the setting of Table 4. We compare the self-supervised and supervised models.

| Model | OA-Crop | Test dataset | | | | | | |
|---|---|---|---|---|---|---|---|---|
| | | COCO-Crop | CIFAR10 | CIFAR100 | CUB | Flowers | Food | Pets |
| MoCov2 | - | 74.30 | 77.58 | 53.26 | 22.90 | 72.09 | 59.70 | 59.25 |
| BYOL | - | 73.36 | 76.62 | 51.79 | 21.95 | 73.77 | 59.49 | 60.72 |
| MoCov2 | CAM | 76.37 | 84.10 | 62.72 | 25.46 | 77.33 | 62.01 | 60.97 |
| BYOL | CAM | 74.92 | 82.79 | 61.13 | 24.34 | 77.83 | 61.83 | 61.27 |
| MoCov2 | GT | 76.44 | 84.03 | 62.81 | 22.59 | 75.09 | 57.47 | 57.67 |
| BYOL | GT | 80.69 | 85.92 | 65.06 | 28.68 | 77.95 | 64.63 | 65.69 |
| Faster R-CNN | - | 81.25 | 88.92 | 67.79 | 30.77 | 77.59 | 66.82 | 66.07 |
| Mask R-CNN | - | 81.45 | 88.59 | 67.72 | 29.05 | 77.57 | 66.27 | 64.40 |

Table 16: Mean AP (%) of MoCov2 and BYOL fine-tuned on the COCO detection and segmentation tasks, following the setting of the table above, using the ResNet-50 architecture.

| | MoCov2 | | | BYOL | | | Faster R-CNN | Mask R-CNN |
|---|---|---|---|---|---|---|---|---|
| | - | CAM | GT | - | CAM | GT | | |
| COCO Detection | 36.3 | 36.6 | 35.7 | 35.1 | 35.6 | 35.1 | 37.0 | 37.9 |
| COCO Segmentation | 32.0 | 32.4 | 31.5 | 31.1 | 31.4 | 31.1 | - | 34.6 |

---

[11]https://pytorch.org/vision/stable/models.html

## D.4 Class-wise accuracy on CIFAR-10

We check if our debiased models suffer from the over-reliance on conspicuous objects as concerned in the potential negative effects section. Table 17 shows the class-wise accuracy of the original and our debiased models on CIFAR-10. OA-Crop does not degrade the performance on certain classes, implying that the concerned bias issue does not occur for our considered transfer scenario.

Table 17: Class-wise linear evaluation (%) of MoCov2 and BYOL on CIFAR-10 under the ResNet-50 architecture, following the setting of Table 4. OA-Crop is not biased to the certain classes.

| Model | OA-Crop | Airplane | Automobile | Bird | Cat | Deer | Dog | Frog | Horse | Ship | Truck |
|---|---|---|---|---|---|---|---|---|---|---|---|
| MoCov2 | - | 79.5 | 86.9 | 67.2 | 61.6 | 74.0 | 68.1 | 82.1 | 79.2 | 89.4 | 87.8 |
| MoCov2 | CAM | 88.9 (+9.4) | 92.8 (+5.9) | 74.8 (+7.6) | 70.0 (+8.4) | 78.6 (+4.6) | 76.6 (+8.5) | 88.6 (+6.5) | 85.1 (+5.9) | 93.8 (+4.4) | 91.8 (+4.0) |
| MoCov2 | GT | 89.0 (+9.5) | 94.0 (+7.1) | 76.5 (+9.3) | 70.9 (+9.3) | 78.2 (+4.2) | 73.9 (+5.8) | 87.0 (+4.9) | 86.5 (+7.3) | 92.5 (+3.1) | 91.8 (+4.0) |
| BYOL | - | 78.6 | 87.8 | 64.1 | 62.3 | 68.8 | 68.4 | 83.8 | 78.7 | 87.2 | 86.8 |
| BYOL | CAM | 86.2 (+7.6) | 93.2 (+5.4) | 73.4 (+9.3) | 69.4 (+7.1) | 76.2 (+7.4) | 74.7 (+6.3) | 87.5 (+3.7) | 83.6 (+4.9) | 92.1 (+4.9) | 91.6 (+4.8) |
| BYOL | GT | 90.2 (+11.6) | 94.6 (+6.8) | 76.7 (+12.6) | 73.0 (+10.7) | 80.1 (+11.3) | 79.8 (+11.4) | 89.4 (+5.6) | 88.6 (+9.9) | 93.6 (+6.4) | 93.2 (+6.4) |

## D.5 Comparison with the ImageNet-trained models

We compare the models trained under the COCO dataset (original or with OA-Crop) with the 10% subset of the ImageNet dataset (i.e., ImageNet 10%) under the ResNet-18 architecture in Table 18. We randomly choose 10% of samples to make a similar size (∼100,000) with the COCO dataset. While the models trained under COCO performing better on the COCO-Crop, ImageNet significantly outperforms the other datasets, implying ImageNet has fewer distribution shifts with them.

Table 18: Linear evaluation (%) of MoCov2 and BYOL on various image classification tasks under the ResNet-18 architecture, following the setting of Table 4. We additionally compare with the models trained under the 10% subset of the ImageNet dataset (i.e., ImageNet 10%).

| Model | Dataset | OA-Crop | Test dataset | | | | | | |
|---|---|---|---|---|---|---|---|---|---|
| | | | COCO-Crop | CIFAR10 | CIFAR100 | CUB | Flowers | Food | Pets |
| MoCov2 | COCO | - | 67.38 | 66.83 | 41.85 | 15.36 | 58.81 | 45.88 | 45.37 |
| MoCov2 | COCO | CAM | 69.92 | 76.73 | 53.25 | 16.26 | 64.77 | 48.56 | 47.37 |
| MoCov2 | COCO | GT | 71.60 | 77.99 | 53.32 | 18.19 | 65.43 | 46.41 | 48.68 |
| MoCov2 | ImageNet 10% | - | 66.28 | 75.28 | 48.64 | 23.75 | 67.99 | 48.70 | 64.16 |
| BYOL | COCO | - | 67.74 | 67.82 | 41.96 | 17.24 | 64.79 | 49.58 | 52.90 |
| BYOL | COCO | CAM | 70.85 | 77.37 | 54.79 | 18.24 | 70.56 | 53.16 | 54.27 |
| BYOL | COCO | GT | 76.59 | 81.23 | 58.11 | 22.99 | 73.25 | 55.33 | 59.80 |
| BYOL | ImageNet 10% | - | 68.96 | 78.51 | 55.40 | 29.89 | 78.14 | 55.10 | 70.16 |

## D.6 Second iteration using the CAM from the debiased models

We compare the models trained with the ContraCAM inferred from the original models (Iter. 1) and the debiased models (Iter. 2) in Table 19. Using the debiased models has no additional gain from the original models. Thus, we use the single iteration version for all experiments.

Table 19: Linear evaluation (%) of MoCov2 and BYOL on various image classification tasks under the ResNet-18 architecture, following the setting of Table 4. We compare the models trained with the ContraCAM inferred from the original models (Iter. 1) and debiased models (Iter. 2).

| Model | OA-Crop | Iter. | Test dataset | | | | | | |
|---|---|---|---|---|---|---|---|---|---|
| | | | COCO-Crop | CIFAR10 | CIFAR100 | CUB | Flowers | Food | Pets |
| MoCov2 | - | - | 67.38 | 66.83 | 41.85 | 15.36 | 58.81 | 45.88 | 45.37 |
| MoCov2 | CAM | 1 | 69.92 (+2.54) | 76.73 (+9.90) | 53.25 (+11.40) | 16.26 (+0.90) | 64.77 (+5.96) | 48.56 (+2.68) | 47.37 (+2.00) |
| MoCov2 | CAM | 2 | 68.53 (+1.15) | 76.54 (+9.71) | 52.64 (+10.79) | 16.62 (+1.26) | 64.89 (+6.08) | 47.34 (+1.46) | 46.77 (+1.40) |
| BYOL | - | - | 67.74 | 67.82 | 41.96 | 17.24 | 64.79 | 49.58 | 52.90 |
| BYOL | CAM | 1 | 70.85 (+3.11) | 77.37 (+9.55) | 54.79 (+12.83) | 18.24 (+1.00) | 70.56 (+5.77) | 53.16 (+3.58) | 54.27 (+1.37) |
| BYOL | CAM | 2 | 70.96 (+3.22) | 77.62 (+9.80) | 54.78 (+12.82) | 20.14 (+2.90) | 71.31 (+6.52) | 53.38 (+3.80) | 53.50 (+0.60) |

# E    Additional background bias results

## E.1    Comparison with the copy-and-paste augmentation

We compare the background mixup using the ContraCAM (BG-Mixup) with the copy-and-paste augmentation using the binarized CAM (BG-HardMix) in Table 20. BG-Mixup (CAM) shows better accuracy (e.g., ORIGINAL) and better generalization (e.g., MIXED-RAND) than the BG-HardMix, implying that the soft blending of foreground and background images performs better than the hard copy-and-paste. Indeed, one should consider the confidence of the predicted CAM masks as they are inaccurate. Also, the soft blending gives a further regularization effect of mixup [41].

Table 20: Test accuracy (%) on background shifts following the setting of Table 7. We compare the background mixup using the ContraCAM (BG-Mixup) and the copy-and-paste version using the binarized CAM (BG-HardMix). Bold denotes the best results among the same model.

| Dataset | MoCov2 | | | BYOL | | |
|---|---|---|---|---|---|---|
| | Baseline | BG-Mixup (CAM) | BG-HardMix (CAM) | Baseline | BG-Mixup (CAM) | BG-HardMix (CAM) |
| Original ↑ | 89.17±0.49 | **90.73**±0.05 (+1.56) | 88.96±0.50 (-0.21) | 87.30±0.61 | **89.30**±0.02 (+2.00) | 88.71±0.28 (+1.41) |
| Only-BG-B ↓ | 31.29±2.46 | **29.60**±0.89 (-1.69) | 31.28±2.03 (-0.01) | 25.59±0.78 | 25.70±3.46 (+0.11) | 26.67±1.32 (+1.08) |
| Only-BG-T ↓ | 44.91±0.16 | **41.95**±0.38 (-2.96) | 44.36±1.40 (-0.55) | 42.83±0.51 | **39.94**±0.52 (-2.89) | 42.02±0.45 (-0.81) |
| Only-FG ↑ | 63.62±4.71 | **70.55**±1.71 (+6.93) | 67.75±0.89 (+4.13) | 61.04±0.94 | **67.53**±0.30 (+6.49) | 63.61±1.65 (+2.57) |
| Mixed-Same ↑ | 80.98±0.34 | **84.13**±0.33 (+3.15) | 82.19±0.61 (+1.21) | 79.30±0.31 | **81.28**±0.53 (+1.98) | 81.25±0.48 (+1.95) |
| Mixed-Rand ↑ | 60.34±0.66 | **66.89**±0.54 (+6.55) | 63.46±0.84 (+3.12) | 58.03±0.85 | **63.83**±0.53 (+5.80) | 61.93±0.17 (+3.90) |
| Mixed-Next ↑ | 55.50±0.71 | **63.64**±0.41 (+8.14) | 59.19±0.94 (+3.69) | 53.35±0.36 | **63.05**±3.54 (+9.70) | 58.13±1.22 (+4.78) |
| BG-Gap ↓ | 20.64±0.36 | **17.24**±0.31 (-3.40) | 18.73±0.54 (-1.91) | 21.27±0.64 | **17.45**±0.15 (-3.82) | 19.32±0.42 (-1.95) |

## E.2    Ablation study on the mixup probability

We study the effect of the mixup probability $p_{\texttt{mix}}$, a probability of applying BG-Mixup augmentation. Table 21 shows the BG-Mixup results with varying $p_{\texttt{mix}} \in \{0.2, 0.3, 0.4, 0.5\}$ applied on MoCov2 and BYOL. We first remark that BG-Mixup gives a consistent gain regardless of $p_{\texttt{mix}}$. Despite of the insensitivity on the hyperparameter $p_{\texttt{mix}}$, we choose $p_{\texttt{mix}} = 0.4$ for MoCov2 and $p_{\texttt{mix}} = 0.3$ for BYOL since they performed best for the most datasets in Background Challenge. MoCov2 permits the higher mixup probability since finding the closest sample from the finite batch (i.e., contrastive learning) is easier than clustering infinitely many samples (i.e., positive-only methods).

Table 21: Test accuracy (%) on background shifts following the setting of Table 7. We study the effect of the mixup probability $p_{\texttt{mix}}$. Bold denotes the best results among the same model.

| Model | $p_{\texttt{mix}}$ | Test dataset | | | | | | | |
|---|---|---|---|---|---|---|---|---|---|
| | | Original ↑ | Only-BG-B ↓ | Only-BG-T ↓ | Only-FG ↑ | Mixed-Same ↑ | Mixed-Rand ↑ | Mixed-Next ↑ | BG-Gap ↓ |
| MoCov2 | 0.0 | 88.67 | 28.47 | 44.99 | 58.25 | 81.21 | 60.57 | 56.22 | 20.64 |
| MoCov2 | 0.2 | 90.52 (+1.85) | **26.32** (-2.15) | 43.73 (-1.26) | 69.01 (+10.76) | 82.91 (+1.70) | 64.89 (+4.32) | 62.12 (+5.90) | 18.02 (-2.62) |
| MoCov2 | 0.3 | **91.09** (+2.42) | 31.53 (+3.06) | 42.91 (-2.08) | 68.42 (+10.17) | 84.25 (+3.04) | 66.72 (+6.15) | 63.68 (+7.46) | 17.53 (-3.11) |
| MoCov2 | 0.4 | 90.72 (+2.05) | 28.69 (+0.22) | **42.05** (-2.94) | **72.35** (+14.10) | **84.42** (+3.21) | 67.51 (+6.94) | 64.04 (+7.82) | **16.91** (-3.73) |
| MoCov2 | 0.5 | 90.81 (+2.14) | 30.07 (+1.60) | 42.20 (-2.79) | 68.96 (+10.71) | 83.26 (+2.05) | 66.27 (+5.70) | 64.15 (+7.93) | 16.99 (-3.65) |
| BYOL | 0.0 | 86.72 | 26.32 | 43.41 | 60.22 | 78.96 | 57.11 | 53.01 | 21.85 |
| BYOL | 0.2 | 88.40 (+1.68) | 22.72 (-3.60) | 41.38 (-2.03) | 63.78 (+3.56) | **81.53** (+2.57) | 62.86 (+5.75) | 59.09 (+6.08) | 18.67 (-3.18) |
| BYOL | 0.3 | **89.31** (+2.59) | **22.47** (-3.85) | **39.68** (-3.73) | 67.36 (+7.14) | 80.89 (+1.93) | 63.58 (+6.47) | **61.01** (+8.00) | **17.31** (-4.54) |
| BYOL | 0.4 | 88.47 (+1.75) | 29.04 (+2.72) | 40.77 (-2.64) | 66.20 (+5.98) | 81.33 (+2.37) | 62.77 (+5.66) | 59.09 (+6.08) | 18.56 (-3.29) |
| BYOL | 0.5 | 88.02 (+1.30) | 29.48 (+3.16) | 40.20 (-3.21) | **69.09** (+8.87) | **81.53** (+2.57) | 62.40 (+5.29) | 59.04 (+6.03) | 19.13 (-2.72) |

## E.3 ContraCAM vs. GT masks on the distribution shifts

We provide distribution shift results of the copy-and-paste augmentation using ground-truth masks (BG-HardMix (GT)) in Table 22. BG-HardMix also improves the performance on distribution shifts by enforcing object-centric learning, but BG-Mixup performs better due to both object-centricness and input interpolation. Recall that BG-HardMix (GT) uses ground-truth masks and thus performs better for background shifts; yet, BG-Mixup is better for the distribution shifts.

Table 22: Test accuracy (%) on distribution shifts following the setting of Table 8. We compare the copy-and-paste version using the ground-truth masks (BG-HardMix (GT)) and the background mixup using the ContraCAM masks (BG-Mixup (CAM)). Bold denotes the best results.

| Model | Augmentation | Test dataset | | | |
|---|---|---|---|---|---|
| | | ImageNet-Sketch-9 | Stylized-ImageNet-9 | ImageNet-R-9 | ImageNet-C-9 |
| MoCov2 | Baseline | 46.70±0.67 | 25.66±0.54 | 37.51±0.80 | 31.82±0.40 |
| MoCov2 | +BG-Mixup (CAM) | **52.15**±0.93 (+5.45) | **33.36**±0.61 (+7.70) | **41.50**±0.45 (+3.99) | **44.39**±0.89 (+12.57) |
| MoCov2 | +BG-HardMix (GT) | 51.60±0.91 (+4.90) | 29.95±2.64 (+4.09) | 40.15±0.34 (+3.39) | 31.45±1.20 (-0.37) |
| BYOL | Baseline | 45.15±1.12 | 23.80±0.45 | 36.21±0.31 | 28.62±0.06 |
| BYOL | +BG-Mixup (CAM) | **52.40**±0.70 (+7.25) | **27.01**±0.74 (+3.21) | 39.62±0.21 (+3.41) | **33.83**±0.28 (+5.21) |
| BYOL | +BG-HardMix (GT) | 51.57±1.68 (+6.42) | 26.72±0.38 (+2.92) | **40.09**±0.41 (+3.88) | 31.04±0.52 (+2.42) |

## E.4 Mixup and CutMix on the background shifts

We provide background shift results of Mixup and CutMix in Table 23. Since they are not designed for addressing the background bias, they are not effective on the Background Challenge benchmarks. In contrast, the background mixup is effective on both background and distribution shifts.

Table 23: Test accuracy (%) on background shifts following the setting of Table 7. We additionally compare with the Mixup and CutMix. Bold denotes the best results.

| Model | Augmentation | Test dataset | | | | | | | |
|---|---|---|---|---|---|---|---|---|---|
| | | Original ↑ | Only-BG-B ↓ | Only-BG-T ↓ | Only-FG ↑ | Mixed-Same ↑ | Mixed-Rand ↑ | Mixed-Next ↑ | BG-Gap ↓ |
| MoCov2 | Baseline | 89.17 | 31.29 | 44.91 | 63.62 | 80.98 | 60.34 | 55.50 | 20.64 |
| MoCov2 | +Mixup [41] | 88.51 (-0.66) | **28.54** (-2.75) | 44.41 (-0.50) | 69.53 (+5.91) | 80.85 (-0.13) | 60.68 (+0.34) | 57.25 (+1.75) | 20.17 (-0.47) |
| MoCov2 | +CutMix [42] | 88.72 (-0.45) | 32.47 (+1.18) | 47.99 (+3.08) | 63.76 (+0.14) | 81.48 (+0.50) | 59.05 (-1.29) | 53.75 (-1.75) | 22.43 (+1.79) |
| MoCov2 | +BG-Mixup (ours) | **90.73** (+1.56) | 29.60 (-1.69) | **41.95** (-2.96) | **70.55** (+6.93) | **84.13** (+3.15) | **66.89** (+6.55) | **63.64** (+8.14) | **17.24** (-3.40) |
| BYOL | Baseline | 87.30 | **25.59** | 42.83 | 61.04 | 79.3 | 58.03 | 53.35 | 21.27 |
| BYOL | +Mixup [41] | 85.70 (-1.60) | 25.95 (+0.36) | 41.00 (-1.83) | 61.79 (+0.75) | 78.61 (-0.69) | 56.27 (-1.76) | 51.75 (-1.60) | 22.34 (+1.07) |
| BYOL | +CutMix [42] | 86.52 (-0.78) | 28.52 (+2.93) | 45.88 (+3.05) | 61.30 (+0.26) | 79.74 (+0.44) | 56.46 (-1.57) | 51.44 (-1.91) | 23.28 (+2.01) |
| BYOL | +BG-Mixup (ours) | **89.30** (+2.00) | 25.70 (+0.11) | **39.94** (-2.89) | **67.53** (+6.49) | **81.28** (+1.98) | **63.83** (+5.80) | **63.05** (+9.70) | **17.45** (-3.82) |

## E.5 Corruption-wise results on ImageNet-C-9

We provide the corruption-wise results on the ImageNet-C-9 dataset in Table 24. Background mixup using the ContraCAM masks (BG-Mixup (CAM)) shows the overall best performance. Especially, the BG-Mixup performs well for the 'weather' and 'digital' class, e.g., improves 24.41% of the baseline to 54.30% (+29.89%), while less performs for the 'noise' class. Indeed, the 'weather' and 'digital' classes require more understanding of the objects (i.e., shape bias) than the 'noise' class.

Table 24: Corruption-wise test accuracy (%) on the ImageNet-C-9 dataset. Bold denotes the best results.

| Model | Augmentation | Noise | | | Blur | | | | Weather | | | | Digital | | | |
|---|---|---|---|---|---|---|---|---|---|---|---|---|---|---|---|---|
| | | Gaussian | Shot | Impulse | Defocus | Glass | Motion | Zoom | Snow | Frost | Fog | Brightness | Contrast | Elastic | Pixel | JPEG |
| MoCov2 | Baseline | 9.24±1.90 | 9.66±1.65 | 9.11±1.56 | 19.32±1.97 | 19.63±1.55 | 29.92±1.10 | 46.42±1.16 | 43.56±1.98 | 29.94±0.73 | 24.41±1.80 | 72.43±1.80 | 16.39±1.80 | 63.64±1.02 | 25.19±1.82 | 58.37±1.70 |
| MoCov2 | Mixup [41] | 18.59±4.42 | 16.62±3.74 | 17.9±5.69 | 19.63±3.76 | 23.94±6.62 | **37.17±2.39** | **50.48±1.44** | 49.10±2.87 | 45.11±3.52 | 32.28±5.57 | 76.18±0.71 | 27.10±4.12 | 65.55±0.36 | 34.39±3.36 | **63.59±1.51** |
| MoCov2 | CutMix [42] | 8.52±3.04 | 8.51±2.89 | 8.53±3.01 | 17.97±5.86 | 19.12±1.97 | 32.45±3.09 | 45.51±1.11 | 44.82±3.56 | 30.93±0.51 | 26.86±1.47 | 73.24±0.05 | 16.11±3.15 | 64.64±2.97 | 33.85±4.16 | 53.31±1.87 |
| MoCov2 | BG-Mixup (CAM) | **31.86±6.58** | **25.19±4.03** | **31.42±7.82** | **20.13±2.05** | **24.89±3.99** | 35.92±1.79 | 49.54±0.36 | **50.45±1.49** | **50.56±1.63** | **54.30±2.35** | **78.35±0.93** | **35.76±4.88** | **65.85±1.16** | **49.37±4.00** | 62.25±1.12 |
| MoCov2 | BG-HardMix (GT) | 10.42±1.41 | 12.54±1.92 | 10.27±1.24 | 16.90±0.97 | 21.49±4.85 | 31.36±2.50 | 44.26±1.47 | 44.08±0.49 | 29.19±1.34 | 27.62±3.02 | 74.18±0.54 | 14.03±1.05 | 64.81±1.37 | 21.28±1.12 | 49.41±0.53 |
| BYOL | Baseline | 6.50±0.50 | 7.27±0.78 | 6.51±0.55 | 12.54±1.47 | 18.23±0.45 | 27.54±3.35 | 41.57±0.52 | 36.67±0.69 | 26.89±1.87 | 24.42±0.76 | 69.59±0.56 | 14.53±0.35 | 60.55±0.94 | 20.65±0.72 | 55.82±1.87 |
| BYOL | Mixup [41] | **14.73±4.59** | **13.85±6.30** | **13.39±3.57** | 12.41±1.29 | 18.48±9.29 | 26.17±3.00 | 38.52±1.27 | **47.55±0.48** | 34.23±1.39 | 29.18±3.15 | 71.33±0.98 | 12.02±2.44 | 62.25±0.58 | 34.39±3.36 | 58.73±1.12 |
| BYOL | CutMix [42] | 7.01±0.55 | 7.46±0.64 | 7.26±0.68 | **12.88±0.31** | 22.27±2.61 | **28.71±3.56** | 42.17±2.75 | 40.85±0.76 | 27.24±0.95 | 26.67±3.33 | 69.30±0.62 | 15.03±1.10 | 62.24±0.37 | 21.89±1.13 | 54.17±2.20 |
| BYOL | BG-Mixup (CAM) | 8.39±1.57 | 9.20±0.73 | 7.86±1.76 | 12.37±0.05 | 18.55±1.62 | 26.99±2.97 | **43.17±2.96** | 41.48±0.47 | **35.26±2.23** | **43.37±2.23** | 73.24±0.20 | **20.58±0.77** | 60.92±0.45 | **41.51±5.35** | **64.59±0.58** |
| BYOL | BG-HardMix (GT) | 8.20±0.15 | 9.43±0.93 | 7.74±0.44 | 10.14±1.26 | **22.97±2.96** | 26.86±2.71 | 42.44±1.24 | 41.38±0.29 | 32.01±1.10 | 30.89±1.19 | **76.58±0.70** | 16.17±0.92 | **62.87±0.34** | 18.24±2.87 | 59.76±3.69 |

## E.6 Results on additional datasets

We additionally evaluate the generalization performance of background mixup on ObjectNet [60] and SI-Score [61], datasets for distribution shift and background shift, respectively. Following previous experiment settings, we train a linear classifier on ImageNet-9 and evaluate the 9-superclass subset of ObjectNet and SI-Score, denoted by adding '-9' in suffix. Table 25 shows that background mixup outperforms the vanilla MoCov2/BYOL (and also Mixup and CutMix) for both datasets. Note that BG-Mixup (CAM) performs better than BG-HardMix (GT) for ObjectNet-9 (distribution-shifted) but less effective for SI-Score-9 (background-shifted).

Table 25: Test accuracy (%) of a linear classifier trained on ImageNet-9 and evaluated on additional distribution-shifted and background-shifted datasets, following the setting of Table 7.

| Model | Augmentation | Test dataset | |
|---|---|---|---|
| | | ObjectNet-9 | SI-Score-9 |
| MoCov2 | Baseline | $24.96_{\pm 2.81}$ | $59.04_{\pm 2.33}$ |
| MoCov2 | +Mixup [41] | $28.53_{\pm 0.52}$ (+3.57) | $58.60_{\pm 1.70}$ (–0.44) |
| MoCov2 | +CutMix [42] | $26.87_{\pm 5.22}$ (+1.91) | $56.33_{\pm 1.57}$ (–2.71) |
| MoCov2 | +BG-Mixup (CAM) | $\mathbf{29.28}_{\pm 3.84}$ (+4.32) | $63.50_{\pm 0.52}$ (+4.46) |
| MoCov2 | +BG-HardMix (GT) | $27.48_{\pm 4.30}$ (+2.52) | $\mathbf{68.70}_{\pm 0.48}$ (+9.66) |
| BYOL | Baseline | $25.38_{\pm 1.59}$ | $55.70_{\pm 1.24}$ |
| BYOL | +Mixup [41] | $25.97_{\pm 2.90}$ (+0.59) | $58.15_{\pm 0.78}$ (+2.45) |
| BYOL | +CutMix [42] | $22.22_{\pm 5.02}$ (–3.16) | $55.78_{\pm 1.79}$ (+0.08) |
| BYOL | +BG-Mixup (CAM) | $\mathbf{31.38}_{\pm 0.52}$ (+6.00) | $62.77_{\pm 1.04}$ (+7.07) |
| BYOL | +BG-HardMix (GT) | $30.78_{\pm 4.29}$ (+5.40) | $\mathbf{69.00}_{\pm 0.48}$ (+13.30) |