# OpenReview forum: "Object-aware Contrastive Learning for Debiased Scene Representation"
_NeurIPS.cc/2021/Conference — NeurIPS 2021 Poster_

### Official Review · Reviewer_ba1p · 2021-07-16

**Rating:** 6
**Confidence:** 5

**Summary:**

The paper proposes object-aware contrastive learning to learn the visual representation from unlabeled images by debiasing the spurious scene correlations. The authors claim there are two types of spurious scene correlations, contextual bias and background bias, which harm the generalization of the model. A novel self-supervised object localization method ContraCAM is proposed to find the unbiased pairs of positives and negatives. Lots of experiments are well done and show the advantages of the proposed ContraCAM.

**Limitations And Societal Impact:**

- the authors emphasize the generalization. But the experiments about the validation of the model's generalization are limited. I suggest the authors provide detailed experiments and analysis to demonstrate the advantages of the proposed method on generalization. For example, trained on a dataset while testing on another unseen dataset.
- the authors claim there are two types of scene bias, contextual bias and background bias. but how do those biases affect scene representation? what is the co-occurrence in scene representation?
-if there are multiple types of objects in the same image, how to guarantee the generalization of scene representation?

**Main Review:**

The main contribution of this paper is the ContraCAM, which is developed upon the CAM algorithm and to find the most discriminative regions. As the name of ContraCAM, the authors introduce contrastive scores into the CAM algorithm to find the salient regions or objects iteratively. The proposed solution is novel and effective. Combining the unsupervised object localization and unsupervised representation learning with the self-supervised manner may be effective in scene representation.

**Time Spent Reviewing:**

8

---

> ### Author Response · Authors · 2021-08-10
> **Response to Reviewer ba1p**
>
> Dear Reviewer ba1p,
>
> Thank you for your valuable feedback and comments.
>
> ---
>
> *Additional generalization experiments*
>
> We first remark that the background mixup experiments (Table 5, Table 6) consider the generalization scenario that the classifier is trained on (seen) ImageNet-9 and evaluated on (unseen) background- and distribution-shifted datasets.
>
> Following your suggestion, we further evaluate the generalization performance of our proposed object-aware random crop (OA-Crop) under the above generalization scenario, as shown in the following table:
>
> Generalization performance (%) of object-aware random crop
> \begin{array}{lllllll}
> \text{Model} & \text{Crop} & & & \text{Test dataset} & & \newline
> & & \text{ImageNet-9} & \text{ImageNet-Sketch-9} & \text{Stylized-ImageNet-9} & \text{ImageNet-R-9} & \text{ImageNet-C-9} \newline
> \hline
> \text{MoCov2} & \text{Baseline}             & \text{84.67} & \text{41.44} & \text{18.94} & \text{32.40} & \text{26.08} \newline
> \text{MoCov2} & \text{OA-Crop (CAM)} & \text{84.54 (-0.13)} & \text{43.11 (+1.68)} & \text{20.50 (+1.56)} & \text{32.35 (-0.05)} & \text{27.85 (+1.77)} \newline
> \text{MoCov2} & \text{OA-Crop (GT)}    & \text{82.49 (-2.18)} & \text{46.85 (+5.42)} & \text{22.18 (+3.24)} & \text{33.68 (+1.28)} & \text{26.81 (+0.73)} \newline
> \hline
> \text{BYOL} & \text{Baseline}             & \text{84.07} & \text{44.28} & \text{17.91} & \text{32.13} & \text{27.51} \newline
> \text{BYOL} & \text{OA-Crop (CAM)} & \text{84.67 (+0.60)} & \text{45.05 (+0.77)} & \text{20.21 (+2.29)} & \text{32.64 (+0.51)} & \text{28.70 (+1.19)} \newline
> \text{BYOL} & \text{OA-Crop (GT)}    & \text{86.72 (+2.65)} & \text{51.52 (+7.25)} & \text{22.95 (+5.03)} & \text{36.28 (+4.15)} & \text{31.65 (+4.14)} \newline
> \end{array}
>
> We found that OA-Crop improves the performance on various distribution-shifted datasets: ImageNet-Sketch-9, Stylized-ImageNet-9, and ImageNet-C-9, while it performs on par with the vanilla models on the original ImageNet-9. We will include the results in the revised manuscript.
>
> ---
>
> *Effects of scene biases for representation*
>
> The contextual bias, i.e., co-occurrence of different objects, entangles the representation of different objects (Appendix D.2) and makes it less discriminative for downstream classification tasks (Table 3). The background bias, i.e., co-occurrence of object and background, makes the representation over-rely on the background instead of objects and fails to predict the objects in front of rare backgrounds (Table 5).
>
> We provide an additional analysis that the vanilla MoCov2 suffers from the over-entanglement of different objects, and OA-Crop resolves it, as shown in the following table:
>
> Classification error (%) of co-occurring classes (giraffe vs. zebra) over epochs
> \begin{array}{lcccccccc}
> \text{Epoch} & 100 & 200 & 300 & 400 & 500 & 600 & 700 & 800 \newline
> \hline
> \text{MoCov2}    & 22.5 & 1.9 & 3.8 & 2.7 & 7.7 & 4.6 & 6.6 & 8.5 \newline
> \text{+OA-Crop} & 1.6 & 1.7 & 2.4 & 2.4 & 5.2 & 2.9 & 1.6 & 1.3 \newline
> \end{array}
>
> Specifically, we report the classification error of co-occurring classes (giraffe vs. zebra) over epochs. OA-Crop shows consistent results while the error of vanilla MoCov2 increases as the epochs go, i.e., OA-Crop reduces the over-entanglement and achieves improved discriminability. We will include the results in the revised manuscript.
>
> Experimental details: We compute the one-shot classification error via a nearest neighbor classifier upon the fixed representation, averaging the five trials.
>
> ---
>
> *Multiple types of objects*
>
> Contrastive learning with vanilla random crop attracts the different objects in multi-object images, which harms the discriminability of scene representation. To address this issue, we infer the objects in the scene and restrict the positive pairs to come from the same object. Since our method has a stronger discriminative power, it achieves a better generalization performance. For example, if the needle and thread always co-occur in the training dataset, the vanilla MoCov2/BYOL may not classify the separated ones (in some unseen dataset) while our method can.
>
> ---
>
> Please let us know if you have any further concerns.
>
> Sincerely,
> Authors

---

### Official Review · Reviewer_HNaz · 2021-07-16

**Rating:** 7
**Confidence:** 3

**Summary:**

This paper proposes a new learning scheme for representation learning. Using class activation maps, augmentations are obtained for contrastive learning that are aware of the objects in the image. Results include results on multiple classification, segmentation, and detection datasets.

Note on 10 sept 2021: rebuttal has been read and discussed with colleague reviewers and AC

**Limitations And Societal Impact:**

Section 5 discusses limitations and concerns of the proposed method.

Methodologically, a major limitation is the reliance on labels on the training set. Lines 179-180 outline a protocol that trains 800 epochs on COCO dataset before obtaining the CAM masks. Therefore, the method is not entirely “self-supervised”.


**Main Review:**

## Strengths:

* Uses object-aware signals for representation learning.

* Figure 3 clearly explains the different steps in the generation of training data.

* Table 1, 2, and 3 show results on multiple datasets.

* Table 4 studies a case where crops obtained from the proposed method might even work better than crops based on the ground truth boxes. This conclusion could have a strong impact on future object-aware contrastive learning.

## Weaknesses:


* How much does the proposed method rely on a good initialized model? The class activation maps in equation (2) will only point to an object of interest whenever the model achieves “good enough” discriminability.
In this light, line 181 confuses me. Do I understand correctly a model is fully trained (on a labelled dataset), then provides CAM masks from which a from-scratch model is learned? If so, the results should be compared to a supervised model that uses the same labelled data.

* Table 1 & table 3: given that this method requires labels (line 179-180), it would be fair to compare with a supervised model.

* Table 4: Comparison with a supervised model would be useful in this table. If I understand correctly from line 178, the ContraCAM maps were obtained from a model that trained 800 epochs on the COCO dataset.

## Comments:

* Figure 1B: the message of this diagram is unclear. The blue line might as well be overfitting and possible to mitigate via regularisation. Is the suggestion that the proposed method adds additional learning signals and so is less amenable to overfitting?


* Line 51: Object net [3] also evaluates this kind of co-occurrences.

* Line 74: effect of image background was also studied in [4].

* LIne 100: how does the iterative procedure influence results? How good are results
obtained with just one iteration?

* Line 274: Object-aware self supervised learning was also studied in [1] and [2]




[1] Pirk et al. "Online object representations with contrastive learning." arXiv preprint arXiv:1906.04312 (2019).

[2] Romijnders et al. "Representation learning from videos in-the-wild: An object-centric approach." WACV 2021.

[3] Barbu et al. "Objectnet: A large-scale bias-controlled dataset for pushing the limits of object recognition models." (2019).

[4] Yung et al. "SI-Score: An image dataset for fine-grained analysis of robustness to object location, rotation and size." arXiv (2021)


Edits after rebuttal:
Authors clarified that models are pre-trained with self-supervised learning. (I erroneously interpreted lines 176-181 as supervised pre-training.) This clarification puts the results in a better light and I will increase review score. I *strongly* encourage authors to clarify this paragraph in a subsequent version of the paper.

**Time Spent Reviewing:**

2

---

> ### Author Response · Authors · 2021-08-10
> **Response to Reviewer HNaz**
>
> Dear Reviewer HNaz,
>
> Thank you for your valuable feedback and comments.
>
> ---
>
> **Usage of supervised labels**
>
> Our method does not use any human-annotated supervision at all. Our proposed ContraCAM predicts the CAM masks from a vanilla self-supervised model trained under an unlabelled dataset. Then, we extract the bounding boxes from the CAM masks (smallest rectangle around the mask) as in line 180. Finally, we train a debiased self-supervised model to address the scene biases by utilizing the precomputed CAM masks and boxes.
>
> Nevertheless, we provide the comparisons with the supervised models, which can be of interest for readers:
> - ContraCAM vs. Classifier CAM for object localization (Table 1)
> - Self-supervised vs. Supervised backbones for representation learning on the COCO dataset (Table 3, Table 4)
>
> ---
>
> *Object localization results*
>
> We compare the localization performance of ContraCAM (using MoCov2) and classifier CAM (using a supervised model). We train all models solely from the target dataset and evaluate the performance on the same dataset, as shown in the following table:
>
> Mask mIoU of ContraCAM and classifier CAM
> \begin{array}{lccc}
> \text{Method} & \text{CUB} & \text{Flowers} & \text{ImageNet-9} \newline
> \hline
> \text{ContraCAM}      & 0.460 & 0.776 & 0.427 \newline
> \text{Classifier CAM} & 0.451 & 0.633 & 0.509 \newline
> \end{array}
>
> Interestingly, ContraCAM performs better than the classifier CAM on CUB and Flowers datasets. We conjecture this is because CUB and Flowers have a small number of training samples (i.e., limited supervised signals); the supervised classifier is prone to overfitting. We will include the results in the revised manuscript.
>
> Experimental details: We train the supervised models using  the same hyperparameter setup as MoCov2 (except removing strong contrastive augmentations) for ImageNet-9 but reduce the epochs and slightly modify the setup for CUB and Flowers since they are prone to overfitting. We omit COCO since it is not a classification dataset. The models are well-trained: classification accuracies of CUB, Flowers, and ImageNet-9 are 61%, 60%, 88%, respectively. We apply the iterative version of the classifier CAM with CRF as ContraCAM.
>
> ---
>
> *COCO representation learning results*
>
> We provide the linear evaluation (Table 3) and detection/segmentation (Table 4) results of two popular supervised models: Faster R-CNN and Mask R-CNN, trained via bounding boxes and instance segmentation, respectively. The results are in the following table:
>
> Linear evaluation (%) and mean AP (%) of self-supervised and supervised models
> \begin{array}{lc|ccccccc|cc}
> \text{Model} & \text{OA-Crop} & & & & \text{Classification} & & & & \text{COCO} & \text{COCO} \newline
> & & \text{COCO-Crop} & \text{CIFAR10} & \text{CIFAR100} & \text{CUB} & \text{Flowers} & \text{Food} & \text{Pets} & \text{Detection} & \text{Segmentation} \newline
> \hline
> \text{Self-supervised} \newline
> \hline
> \text{MoCov2} & \text{-}        & 74.30 & 77.58 & 53.26 & 22.90 & 72.09 & 59.70 & 59.25 & 36.3 & 32.0 \newline
> \text{BYOL}     & \text{-}        & 73.36 & 76.62 & 51.79 & 21.95 & 73.77 & 59.49 & 60.72 & 35.1 & 31.1 \newline
> \text{MoCov2} & \text{CAM} & 76.37 & 84.10 & 62.72 & 25.46 & 77.33 & 62.01 & 60.97 & 36.6 & 32.4 \newline
> \text{BYOL}     & \text{CAM} & 74.92 & 82.79 & 61.13 & 24.34 & 77.83 & 61.83 & 61.27 & 35.6 & 31.4 \newline
> \text{MoCov2} & \text{GT}    & 76.44 & 84.03 & 62.81 & 22.59 & 75.09 & 57.47 & 57.67 & 35.7 & 31.5 \newline
> \text{BYOL}     & \text{GT}    & 80.69 & 85.92 & 65.06 & 28.68 & 77.95 & 64.63 & 65.69 & 35.1 & 31.1 \newline
> \hline
> \text{Supervised} \newline
> \hline
> \text{Faster R-CNN} & \text{-} & 81.25 & 88.92 & 67.79 & 30.77 & 77.59 & 66.82 & 66.07 & 37.0 & \text{-} \newline
> \text{Mask R-CNN}   & \text{-} & 81.45 & 88.59 & 67.72 & 29.05 & 77.57 & 66.27 & 64.40 & 37.9 & 34.6 \newline
> \end{array}
>
> The supervised models perform better than self-supervised models, but our proposed object-aware random crop (OA-Crop) significantly reduces their gap, e.g., from 53% to 63% for the CIFAR100 dataset. Also, BYOL trained via OA-Crop using ground-truth boxes almost matches the linear evaluation performance of the supervised models. We will include the results in the revised manuscript.
>
> Experimental details: We use the publicly available TorchVision models trained on the COCO dataset using the ResNet-50 architecture. We used the pretrained weights for detection/segmentation experiments and trained a linear classifier upon the pretrained weights for linear evaluation experiments.
>
> ---
>
> **Other comments**
>
> *Clarification of Figure 1B*
>
> We will revise the caption of Figure 1B. The classification performance of vanilla MoCov2 decreases since it entangles the features of different objects, as shown in Appendix D.2. Since it is a fundamental issue of the random crop, we hypothesize general regularization techniques (e.g., l2-regularization) would not resolve the problem. Instead, we propose an object-aware random crop (OA-Crop) to address the issue.
>
> We provide an additional analysis that the vanilla MoCov2 suffers from the over-entanglement of different objects, and OA-Crop resolves it, as shown in the following table:
>
> Classification error (%) of co-occurring classes (giraffe vs. zebra) over epochs
> \begin{array}{lcccccccc}
> \text{Epoch} & 100 & 200 & 300 & 400 & 500 & 600 & 700 & 800 \newline
> \hline
> \text{MoCov2}    & 22.5 & 1.9 & 3.8 & 2.7 & 7.7 & 4.6 & 6.6 & 8.5 \newline
> \text{+OA-Crop} & 1.6 & 1.7 & 2.4 & 2.4 & 5.2 & 2.9 & 1.6 & 1.3 \newline
> \end{array}
>
> Specifically, we report the classification error of co-occurring classes (giraffe vs. zebra) over epochs. OA-Crop shows consistent results while the error of vanilla MoCov2 increases as the epochs go, i.e., OA-Crop reduces the over-entanglement and achieves improved discriminability. We will include the results in the revised manuscript.
>
> Experimental details: We compute the one-shot classification error via a nearest neighbor classifier upon the fixed representation, averaging the five trials.
>
> ---
>
> *Additional related datasets*
>
> We evaluate the generalization performance of our proposed background mixup (BG-Mixup) on the suggested datasets: ObjectNet [3] and SI-Score [4], as shown in the following table:
>
> Generalization performance (%) of background mixup on additional datasets
> \begin{array}{llll}
> \text{Model} & \text{Augmentation} & \text{Test dataset} & \newline
> & & \text{ObjectNet-9} & \text{SI-Score-9} \newline
> \hline
> \text{MoCov2} & \text{Baseline}                  & \text{24.96} & \text{59.04} \newline
> \text{MoCov2} & \text{+Mixup}                    & \text{28.53 (+3.57)} & \text{58.60 (-0.44)} \newline
> \text{MoCov2} & \text{+CutMix}                  & \text{26.87 (+1.91)} & \text{56.33 (-2.71)} \newline
> \text{MoCov2} & \text{+BG-Mixup (CAM)}  & \text{29.28 (+4.32)} & \text{63.50 (+4.46)} \newline
> \text{MoCov2} & \text{+BG-HardMix (GT)} & \text{27.48 (+2.52)} & \text{68.70 (+9.66)} \newline
> \hline
> \text{BYOL} & \text{Baseline}                  & \text{25.38} & \text{55.70} \newline
> \text{BYOL} & \text{+Mixup}                    & \text{25.97 (+0.59)} & \text{58.15 (+2.45)} \newline
> \text{BYOL} & \text{+CutMix}                  & \text{22.22 (-3.16)} & \text{55.78 (+0.08)} \newline
> \text{BYOL} & \text{+BG-Mixup (CAM)}  & \text{31.38 (+6.00)} & \text{62.77 (+7.07)} \newline
> \text{BYOL} & \text{+BG-HardMix (GT)} & \text{30.78 (+5.40)} & \text{69.00 (+13.30)} \newline
> \end{array}
>
> We observe the consistent results with our paper: +BG-Mixup outperforms the vanilla MoCov2/BYOL and +Mixup/+CutMix for both datasets. Also, note that BG-Mixup (using CAM) outperforms BG-HardMix (GT), a variant of BG-Mixup using ground-truth masks, for distribution-shifted dataset (ObjectNet-9); while less effective for background-shifted dataset (SI-Score-9). We will include the results in the revised manuscript.
>
> Experimental details: Following the setup of Table 6 in our paper, we train a linear classifier upon the learned representations. We then evaluate the performance on the 9-superclass subset of ObjectNet and SI-Score, denoted by adding ‘-9’ at the suffix.
>
> ---
>
> *Effect of iterative procedure of ContraCAM*
>
> For large objects, a single iteration of ContraCAM (without a double expansion trick) performs well. However, applying the double expansion trick to detect small objects only covers a small region of the objects. Here, the iterative procedure benefits. Appendix C.2 shows that a sufficient number of iterations improves the localization performance.
>
> ---
>
> *Additional related works on object-aware self-supervised learning*
>
> Thanks for suggesting related works [1,2]. We note that the goal of [1,2] was to additionally utilize the objects of different temporal frames for video representation learning, while our goal is to address scene biases of image representation learning. Also, we proposed a self-supervised technique to detect objects while [1,2] use the pretrained object detector. We will discuss these differences in the revision.
>
> ---
>
> Please let us know if you have any further concerns.
>
> Sincerely,
> Authors

---

### Official Review · Reviewer_1LkH · 2021-07-17

**Rating:** 7
**Confidence:** 4

**Summary:**

This paper proposes an object-aware contrastive learning framework which can localize the objects in a self-supervised manner (ContraCAM) and then uses these localized objects to learn de-biased representations. The authors motivate the proposed approach through spurious scene correlations between different objects or between objects and the background. The proposed ContraCAM approach is an intuitive extension of CAM to contrastively trained models. ContraCAM can detect multiple discriminative objects in an image using an iterative method.

**Limitations And Societal Impact:**

Yes.

**Main Review:**

The paper proposes an interesting and novel approach which extends Class Activation Maps to the contrastive learning framework. The proposed approach, called ContraCAM, can be used to obtain activation maps for discriminative objects in an image. The authors also propose an iterative extension of ContraCAM which enables localization of multiple objects. The paper further proposes some object-aware augmentations for reducing the contextual biases in contrastive learning methods. The authors show that the proposed approach achieves good performance improvements on an existing contrastive learning method. I believe that the novelty of the proposed approach and the strong results and detailed ablation studies presented by the authors make this a strong paper.

Edit: After reading the other reviews and the authors responses, I will maintain my rating and recommend this paper for acceptance.

**Time Spent Reviewing:**

5

---

> ### Author Response · Authors · 2021-08-10
> **Response to Reviewer 1LkH**
>
> Dear Reviewer 1LkH,
>
> We appreciate your positive comments and a nice summary of our manuscript.
>
> As you highlighted, our proposed object localization method ContraCAM is novel and well-improves the existing contrastive learning frameworks by utilizing it for object-aware augmentations, validated by strong experiments and detailed experiments.
>
> We believe our work would be valuable for object localization (e.g., class activation map) and self-supervised learning researchers.
>
> Sincerely,
> Authors

---

### Decision · Program_Chairs · 2021-09-27

**Decision:**

Accept (Poster)

**Comment:**

All reviewers are positive about this paper. The idea of using CAM to localize the most salient object in an image and use that to reduce background and contextual bias in scene-centric (as opposed to object-centered) datasets for selfsupervised learning is interesting and seems to indeed alleviate the issues. I recommend to accept the paper as a poster.